# Delayed gut microbiota maturation in the first year of life is a hallmark of pediatric allergic disease

Courtney Hoskinson [1,2], Darlene L. Y. Dai [1], Kate L. Del Bel[1], Allan B. Becker[3], Theo J. Moraes[4], Piushkumar J. Mandhane[5], B. Brett Finlay[2,6,7], Elinor Simons[3], Anita L. Kozyrskyj[5], Meghan B. Azad[3,8,9], Padmaja Subbarao [4,10,11], Charisse Petersen [1,12] & Stuart E. Turvey[1,12] ✉

Allergic diseases affect millions of people worldwide. An increase in their prevalence has been associated with alterations in the gut microbiome, i.e., the microorganisms and their genes within the gastrointestinal tract. Maturation of the infant immune system and gut microbiota occur in parallel; thus, the conformation of the microbiome may determine if tolerant immune programming arises within the infant. Here we show, using deeply phenotyped participants in the CHILD birth cohort ($n = 1115$), that there are early-life influences and microbiome features which are uniformly associated with four distinct allergic diagnoses at 5 years: atopic dermatitis (AD, $n = 367$), asthma (As, $n = 165$), food allergy (FA, $n = 136$), and allergic rhinitis (AR, $n = 187$). In a subset with shotgun metagenomic and metabolomic profiling ($n = 589$), we discover that impaired 1-year microbiota maturation may be universal to pediatric allergies (AD $p = 0.000014$; As $p = 0.0073$; FA $p = 0.00083$; and AR $p = 0.0021$). Extending this, we find a core set of functional and metabolic imbalances characterized by compromised mucous integrity, elevated oxidative activity, decreased secondary fermentation, and elevated trace amines, to be a significant mediator between microbiota maturation at age 1 year and allergic diagnoses at age 5 years ($\beta_{indirect} = -2.28$; $p = 0.0020$). Microbiota maturation thus provides a focal point to identify deviations from normative development to predict and prevent allergic disease.

Allergic diseases affect hundreds of millions of children worldwide and continue to increase in prevalence[1–4]. These rising rates have coincided with social and environmental changes that have had an intergenerational impact on the stably colonizing microbes and their collective genes that make up our microbiota and microbiome, respectively[5,6].

Established during infancy, the nascent microbiota's initial expansion and fluctuation are particularly sensitive to external

[1]Department of Pediatrics, BC Children's Hospital, University of British Columbia, Vancouver, BC, Canada. [2]Department of Microbiology and Immunology, University of British Columbia, Vancouver, BC, Canada. [3]Department of Pediatrics and Child Health, University of Manitoba, Winnipeg, MB, Canada. [4]Department of Pediatrics, The Hospital for Sick Children, Toronto, ON, Canada. [5]Department of Pediatrics, University of Alberta, Edmonton, AB, Canada. [6]Michael Smith Laboratories, University of British Columbia, Vancouver, BC, Canada. [7]Department of Biochemistry and Molecular Biology, University of British Columbia, Vancouver, BC, Canada. [8]Department of Food and Human Nutritional Sciences, University of Manitoba, Winnipeg, MB, Canada. [9]Manitoba Interdisciplinary Lactation Centre (MILC), Children's Hospital Research Institute of Manitoba, Winnipeg, MB, Canada. [10]Department of Medicine, McMaster University, Hamilton, ON, Canada. [11]Dalla Lana School of Public Health, University of Toronto, Toronto, ON, Canada. [12]These authors jointly supervised this work: Charisse Petersen, Stuart E. Turvey. ✉e-mail: sturvey@cw.bc.ca

influences before reaching a more stable community. Indeed, many risk factors for allergic diseases, including mode of delivery, diet, urban living, and antibiotic exposure, also influence early microbiota membership and structure[7–10]. While this maturation process usually coincides with the development of healthy immune tolerance, allergic sensitization can emerge in some children during the same period as the microbiota is being established[4,11]. Considering the relationship between external risk factors, infant microbiome maturation, and pediatric immune development, interrogating the early-life microbiome has the potential to empower predictive and therapeutic strategies designed to prevent the development of allergic disease.

Although they are often studied in isolation as distinct, organ-specific clinical diagnoses, asthma, allergic rhinitis (or hay fever), food allergy, and atopic dermatitis (or eczema) can share many common etiological mechanisms characterized by aberrant type-2 inflammatory responses and elevated IgE[11–15]. Supporting their shared biological origins, a predictable series of onset for these disorders has been observed in young children, described collectively as the Allergic March[11,15]. Given this evidence, it is difficult to disentangle the environmental and biological underpinnings of individual allergic diseases, and a collective approach that investigates all four of these common pediatric allergic disorders in parallel is of increasing relevance. Previous studies looking at multiple diseases have noted common microbiome associations in individuals diagnosed with distinct individual allergic diseases[16,17]. These studies have set a strong precedent for identifying shared significant microbial features in individuals currently suffering from diseases. To date, few studies have looked at infant microbial associations with multiple distinct allergic disease outcomes, and most lacked the power of the prospective, longitudinal design of the CHILD cohort[18].

In this study, we evaluated four clinically distinct allergic diseases diagnosed at age 5 years in the large, deeply characterized CHILD cohort study. We used a multi-omics approach to profile infant stool collected at study visits scheduled for ages 3 months and 1 year. We found that delayed infant microbiota maturation was shared across each 5-year allergic diagnosis compared to those with no history of allergic sensitization and that this delay in microbiota maturation preceded the diagnosis of allergic disease. Functional implications of this impairment were also observed in each of the diagnoses, including compromised impact on mucous integrity, elevated oxidation potential, decreased secondary fermentation and butyrate production, and increased biogenic amines within infant's guts. Our findings identify common, host-microbiome mechanisms associated with the development of multiple clinically distinct allergic disorders. Prioritizing preventive strategies and therapeutic intervention to modify these host-microbe interactions during infancy may have lasting benefits for preventing pediatric allergic diseases, which often last a lifetime.

## Results

### Defining the epidemiology of allergic diseases
The longitudinal and comprehensive nature of the CHILD study provided a powerful opportunity to clearly define any history of allergic sensitization and current diagnosis of asthma, food allergy, atopic dermatitis, and/or allergic rhinitis (Fig. 1a). To avoid spurious associations stemming from co-occurring or transient allergic conditions, we incorporated only clinical evaluations of participants who had extensive and complete clinical assessments for every study visit from birth to age 5 years ($n = 1115$) (Fig. 1a and Supplementary Table 1). The rigorously defined "healthy" control group comprised 523 children who had no evidence of allergic sensitization at any time in their life (defined as repeatedly negative allergen skin prick tests (SPT), no history of wheezing, and no diagnosis of any of the allergic disorders—atopic dermatitis, asthma, allergic rhinitis, food allergy), as determined by three separate clinical evaluations at 1, 3, and 5 years of age (Fig. 1b). In comparison, 592 children had been diagnosed by an expert physician at the 5-year scheduled visit with one or more allergic disorders

(i.e., atopic dermatitis, asthma, allergic rhinitis, and food allergy), with the majority of these diagnoses (59.4–91.2%) co-occurring with a positive skin prick test (SPT) at one or more of the allergic evaluations (Supplementary Fig. 1a, b).

When we evaluated the association of early-life factors with a diagnosis of allergic disease at age 5 years using a multivariate conditional logistic regression with the study site as strata, the following risk factors emerged: male sex (adjusted odds ratio (aOR): 1.84 [95% CI 1.36, 2.49]; $p = 6.8e{-}05$), history of paternal (aOR: 1.56 [95% CI 1.13, 2.15]; $p = 0.007$) or maternal (aOR: 1.56 [95% CI 1.14, 2.12]; $p = 0.0054$) atopy, and antibiotic usage before age 1 year (aOR: 2.25 [95% CI 1.55, 3.27]; $p = 2.0e{-}05$) (Fig. 2a and Supplementary Table 2). In contrast, any breastfeeding up to age 6 months (aOR: 0.66 [95% CI 0.45, 0.99]; $p = 0.043$) and self-identifying as Caucasian (aOR: 0.44 [95% CI 0.32, 0.61]; $p = 5.1e{-}07$) were negatively associated with an allergy diagnosis (Fig. 2a and Supplementary Table 2). Notably, the significant overlap between these associated risk factors and the diagnosis of individual allergic diseases supports our collective approach to identifying a common etiology within the infant microbiome (Fig. 2b and Supplementary Data 1).

### Delayed infant gut microbiome age associated with allergic diseases at school age
We next evaluated the participants' infant stool microbiomes ($n = 589$ participants) collected at clinical assessments scheduled for 3-month and 1-year visits and quantified via shotgun metagenomic sequencing[19] (Fig. 1b and Supplementary Table 1). Comparing the alpha diversity within the infant microbiome at both timepoints across all the individual allergy diagnoses at age 5 years, we identified a significant decrease in Shannon diversity at age 1 year in infants who went on to have any allergic diagnosis at age 5 years (Fig. 3a, b). Increased diversification is a hallmark of infant gut microbiome dynamics across the first year of life and is accompanied by substantial shifts in microbial abundance[7,20–22]. This process is so linked to early-life development that the composition of the infant microbiota alone can accurately predict an infant's chronological age[23]. To understand whether infant microbiota maturation was ubiquitously associated with school-age allergic diagnoses, we calculated a nested cross-validated, microbiota-derived predicted age using species abundances across the first year of life (Fig. 3c; Pearson $R = 0.89$, $p < 2.2e{-}16$). We then compared microbiota-derived age across each of the four allergic diagnoses at 5 years (Fig. 3d). Infants with no allergic history detected at any time between birth and 5 years had an average microbiota-predicted age of 11.53 (SD 1.32) months at their 1-year visit. In contrast, each of the four allergic diagnoses at age 5 years had a statistically lower predicted microbiota age, despite having the same chronological age (atopic dermatitis $p = 0.000014$; asthma $p = 0.0073$; food allergy $p = 0.00083$; and allergic rhinitis $p = 0.0021$; Fig. 3d, e). Notably, this reduction in microbiota-predicted age at 1 year was detected in children with a 5-year allergy diagnosis, regardless of SPT response history (Supplementary Fig. 1c). Additionally, while a number of children had multiple allergy diagnoses at 5 years, even children with a single allergy diagnosis also had significantly lower predicted age (Supplementary Fig. 1d). Furthermore, microbiota-predicted age remained protective when adjusting for confounding variables (aOR of one or more allergic diagnoses for an IQR increase in microbiota-predicted age: 0.75 [95% CI 0.59, 0.94]; $p = 0.010$) (Supplementary Fig. 2). In summary, reduced microbiota maturation at 1 year of age is associated with an increased risk of being diagnosed with an allergic disease at age 5 years, regardless of the specific allergic condition.

### Shared pathway dysfunction links impaired microbiota maturation and allergic disease development
Given that microbiota maturation impairment at 1 year was present across all individual allergic diagnoses at 5 years, we analyzed the

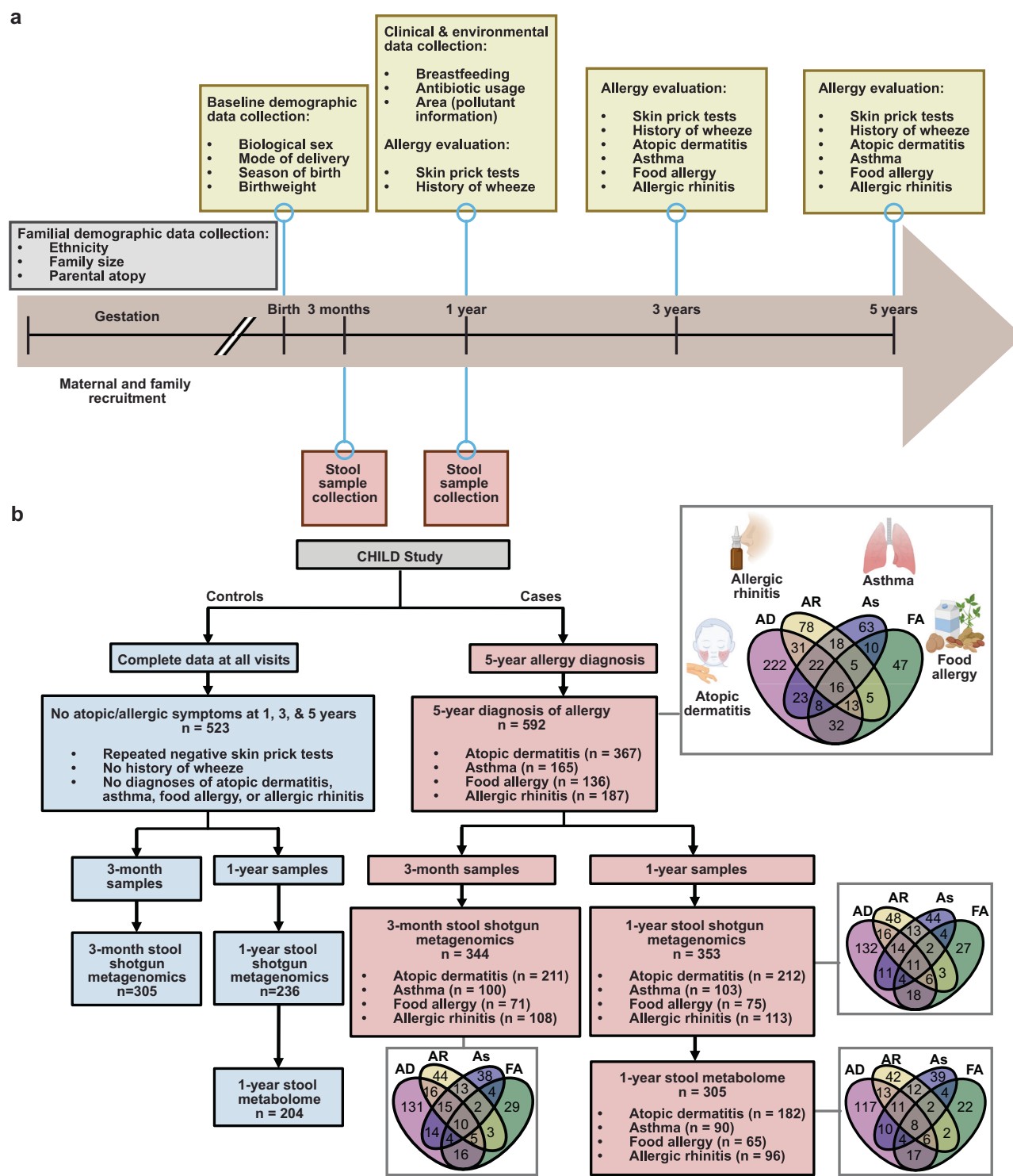

**Fig. 1 | Clinical evaluation of CHILD participants and data collection from biological samples. a** Timeline of CHILD enrollment and clinical evaluations from gestation through 5-year evaluations. **b** Consort diagram of CHILD participants and samples included in this study, including the composition of participant allergic diseases and their interrelated diagnoses. Created with Biorender.com.

underlying taxonomic and functional components associated with microbiota-predicted age and its relationship with allergy development. We first focused on the top 25 species with the highest average importance ranking based on the nested cross-validated random forest model and compared their directional effect on predicted age using a linear mixed-effect model with adjustment for age and a random effect of the sample collection site (Fig. 4a). When we compared species abundance within the 1-year microbiota between children who

did or did not receive an allergic diagnosis at 5 years, we identified 9 overlapping species that were related to microbiota-derived predicted age and showed differential abundance (false-discovery rate (FDR) <0.1) in infants later diagnosed with allergic diseases (Fig. 4b and Supplementary Fig. 3). This included decreases in *Anaerostipes hadrus*, *Fusicatenibacter saccharivorans*, *Eubacterium hallii*, and *Blautia wexlerae* and increases in *Eggerthella lenta*, *Escherichia coli*, *Enterococcus faecalis*, *Clostridium innocuum*, and *Tyzzerella nexilis* in infants who

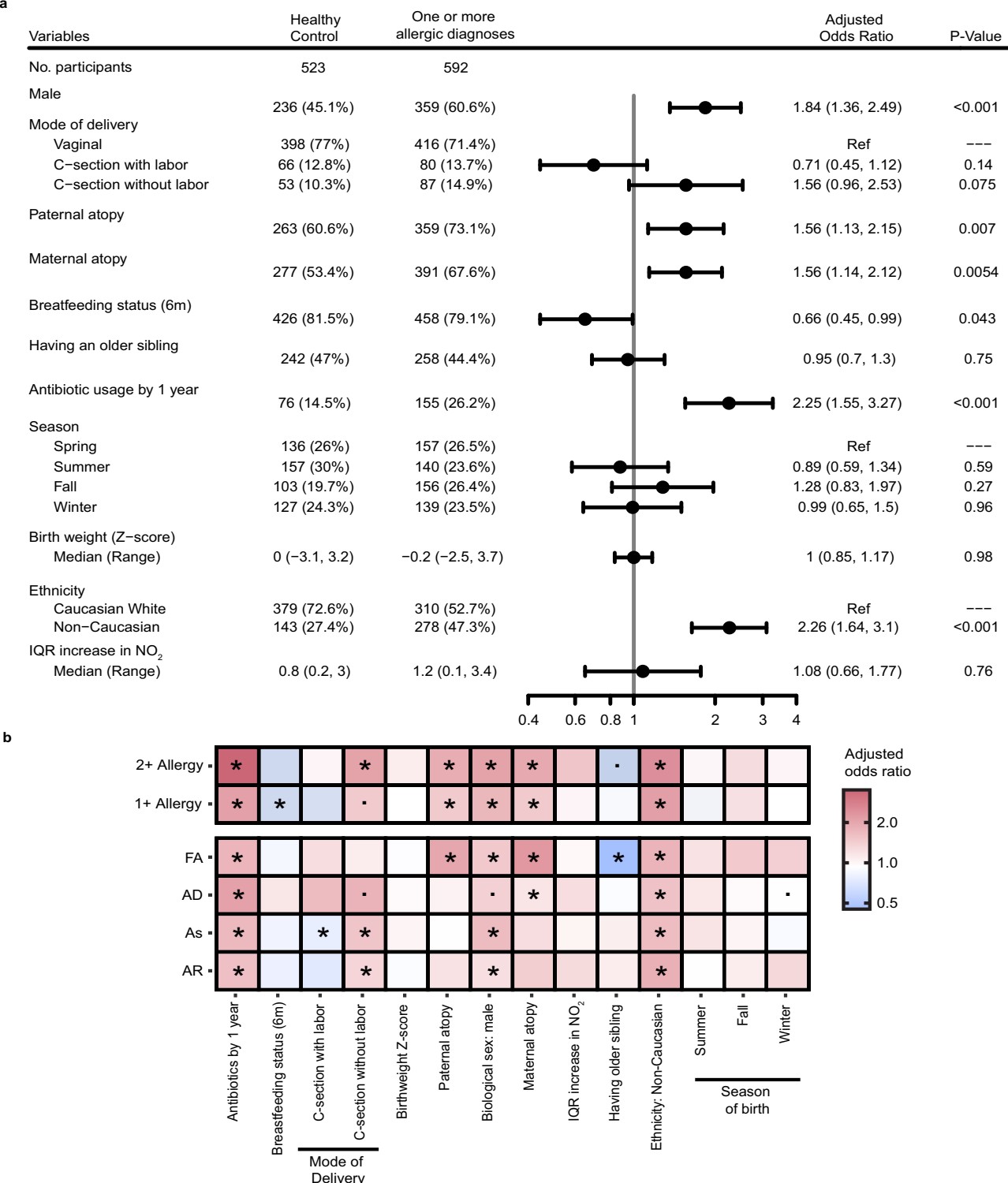

**Fig. 2 | Individual allergic disease progression and influences.** Multivariable conditional logistic regression, using the data collection site as a stratum, evaluating the odds ratio of developing **a** one or more atopic or allergic diagnoses (*n* = 592) and **b** one or more atopic or allergic diagnoses (1+, *n* = 592), two or more diagnoses of (2+, *n* = 107), and each of atopic dermatitis (AD, *n* = 282), food allergy (FA, *n* = 100), asthma (As, *n* = 127), or allergic rhinitis (AR, *n* = 141) when accounting for early-life and familial exposures. (*) *p* < 0.05, (.) *p* < 0.1. For forest plot, data were presented as adjusted odds ratios (95% confidence intervals) and exact *p* values: male *p* = 6.8e-05, antibiotic usage *p* = 2.0e-05, and ethnicity *p* = 5.1e-07.

developed allergic diagnoses by 5 years (Fig. 4c). This pattern, emphasizing the importance of a core group of species, was generally replicated in analyses of individual allergic disorders (Fig. 4c). Within these samples, we further linked increased *C. innocuum* and *T. nexilis* to antibiotic usage, altered *C. innocuum*, *E. lenta*, *E. faecalis*, and *T.* *nexilis* to breastfeeding status at 6 months, and differential *C. innocuum* and *E. lenta* to paternal atopy, identifying some of the environmental and clinical influences that potentially shape the microbiome (Supplementary Table 3). Thus, in addition to altered diversification, changes in the abundance of a core group of species are indicative of

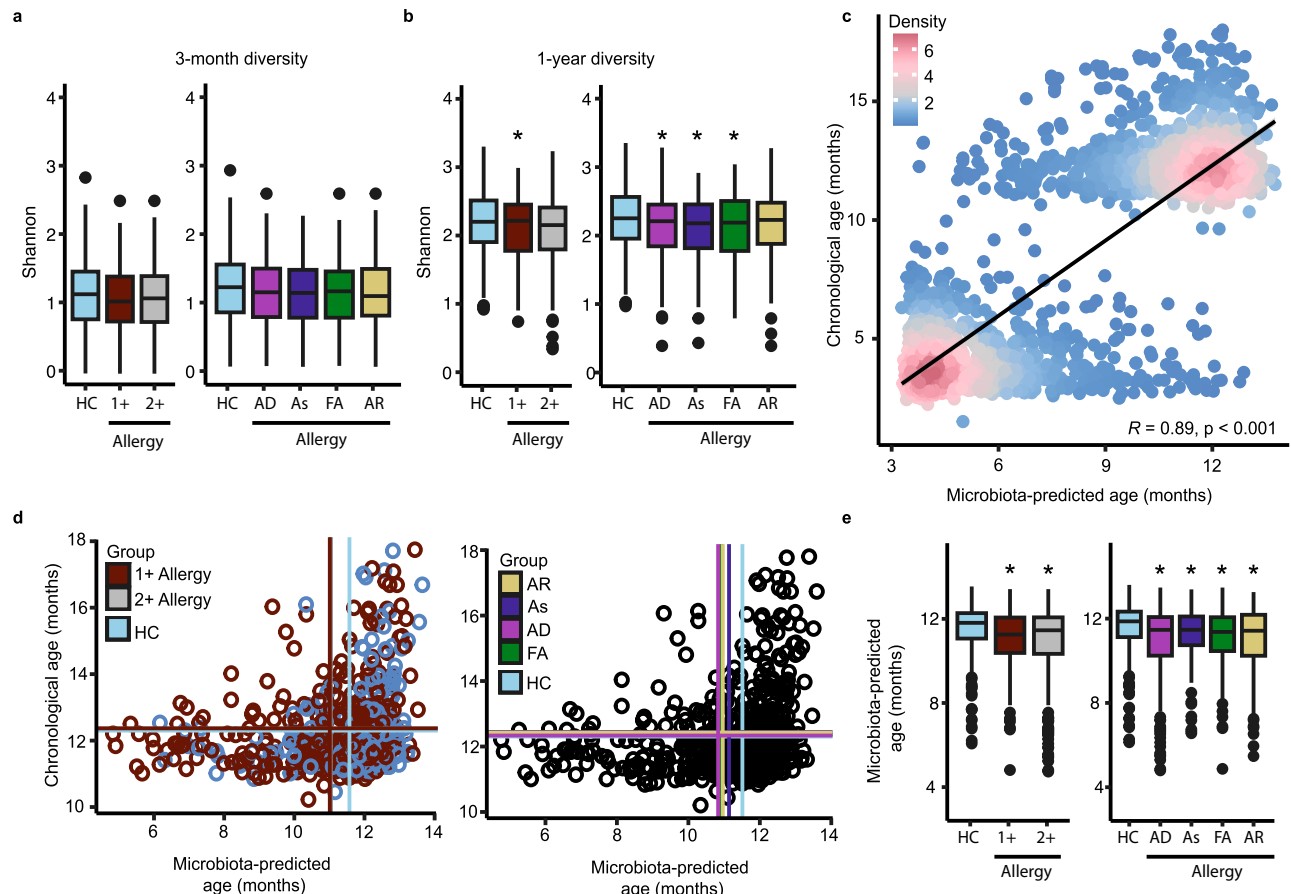

**Fig. 3 | Diversity and microbiome-derived age of the infant's gut.** Shannon diversity index of **a** 3-month samples for one or more atopic or allergic diagnoses (1+, *n* = 344), two or more allergic diagnoses (2+, *n* = 130), and individual clinical diagnoses at 5 years, i.e., atopic dermatitis (AD, *n* = 211), food allergy (FA, *n* = 73), asthma (As, *n* = 100), or allergic rhinitis (AR, *n* = 108), at 5 years, and healthy control (HC, *n* = 244) participants, as well as **b** 1-year samples for 1+ (*n* = 353, *p* = 0.039), 2+ (*n* = 82, *p* = 0.021), and individual clinical diagnoses at 5 years, i.e., AD (*n* = 212, *p* = 0.021), FA (*n* = 75, *p* = 0.043), As (An = 103, *p* = 0.0097), or AR (*n* = 113, *p* = 0.021), at 5 years, and HC (*n* = 236) participants, **c** Scatterplot between chronological age and microbiome-derived age with linear regression line of best fit (Pearson *R* = 0.89, *p* < 2.2e-16).

**d** Predicted age and chronological age for aggregate and individual clinical diagnoses as compared to no diagnoses at 5 years. **e** Predicted age of 1-year samples for 1+ (*n* = 353, *p* = 0.000036), 2+ (*n* = 82, *p* = 0.0023), and individual clinical diagnoses at 5 years, i.e., AD (*n* = 212, *p* = 0.000014), FA (*n* = 75, *p* = 0.00083), As (*n* = 103, *p* = 0.0073), or AR (*n* = 113, *p* = 0.0021), at 5 years, and HC (*n* = 236) participants. *P* values are from Wilcoxon tests between HC and each allergic diagnosis (**b, c, e**). For box plots, data are presented as box plots (center line at the median, upper bound at 75th percentile, lower bound at 25th percentile) with whiskers at minimum and maximum values. (*) *p* < 0.05.

---

both reduced infant gut microbiota maturation and the development of multiple clinically distinct allergic disorders.

Defining the functional impacts of reduced infant gut microbiota maturation could reveal key pathways that might be targeted to prevent the development of persistent allergic disease. This prompted us to perform two multivariable mixed-effect regressions, adjusting for chronological age and using the collection site as a random effect, of the 347 MetaCyc pathways with at least 10% prevalence using the established 5-year allergic diagnoses groups and microbiota-derived predicted age as the respective outcomes in each analysis (Fig. 5a). We identified 193 pathways significantly associated with at least one of the composite or individual 5-year allergic diagnoses and 281 pathways associated with predicted age (FDR <0.1, Supplementary Data 2). Moreover, when we compared MetaCyc pathway abundance between healthy children and those with a 5-year allergic diagnosis, 171 of the 193 significantly altered pathways in allergic groups were also associated with predicted age (Supplementary Data 2).

While we saw similar patterns across all four of the allergy diagnoses, 11 pathways were significantly different in at least two of the allergy diagnoses and one of the composite groups, all of which were also significantly associated with predicted age (Fig. 5b and Supplementary Data 3). Nine of these were negatively associated

with microbiota-predicted age and subsequently elevated in infants who developed allergies. These include pathways corresponding to mucous degradation via cysteine disulfide bond reduction (e.g., Sulfoquinovose degradation I, and molybdopterin biosynthesis[24–26]), increased oxidative respiration (e.g., NAD(P)/NADPH interconversion[26]), and oxidized monosaccharide utilization (e.g., *D*-galactarate and *D*-glucarate degradation[27]). Conversely, two pathways were positively associated with predicted age, as well as protection from allergy development, including methanogenesis from acetate and sulfur oxidation[28,29]. A Spearman correlation analysis revealed a significant connection between *B. wexlerae*, *F. saccharivorans*, *A. hadrus*, and *E. hallii* and pathways with protective associations, while *E. coli* was primarily correlated with pathways that were elevated in infants with a 5-year allergy diagnosis (Fig. 5c). Thus, reduced infant gut microbiota maturation is linked to broad functional dysregulation overlapping with that of the development of allergic disorders.

## Stool metabolomic profile of the 1-year infant gut and its association with key microbiome features
Metabolites within the gut play an essential role in the microbiome's biological impact on the host[30–33]. We next sought to understand

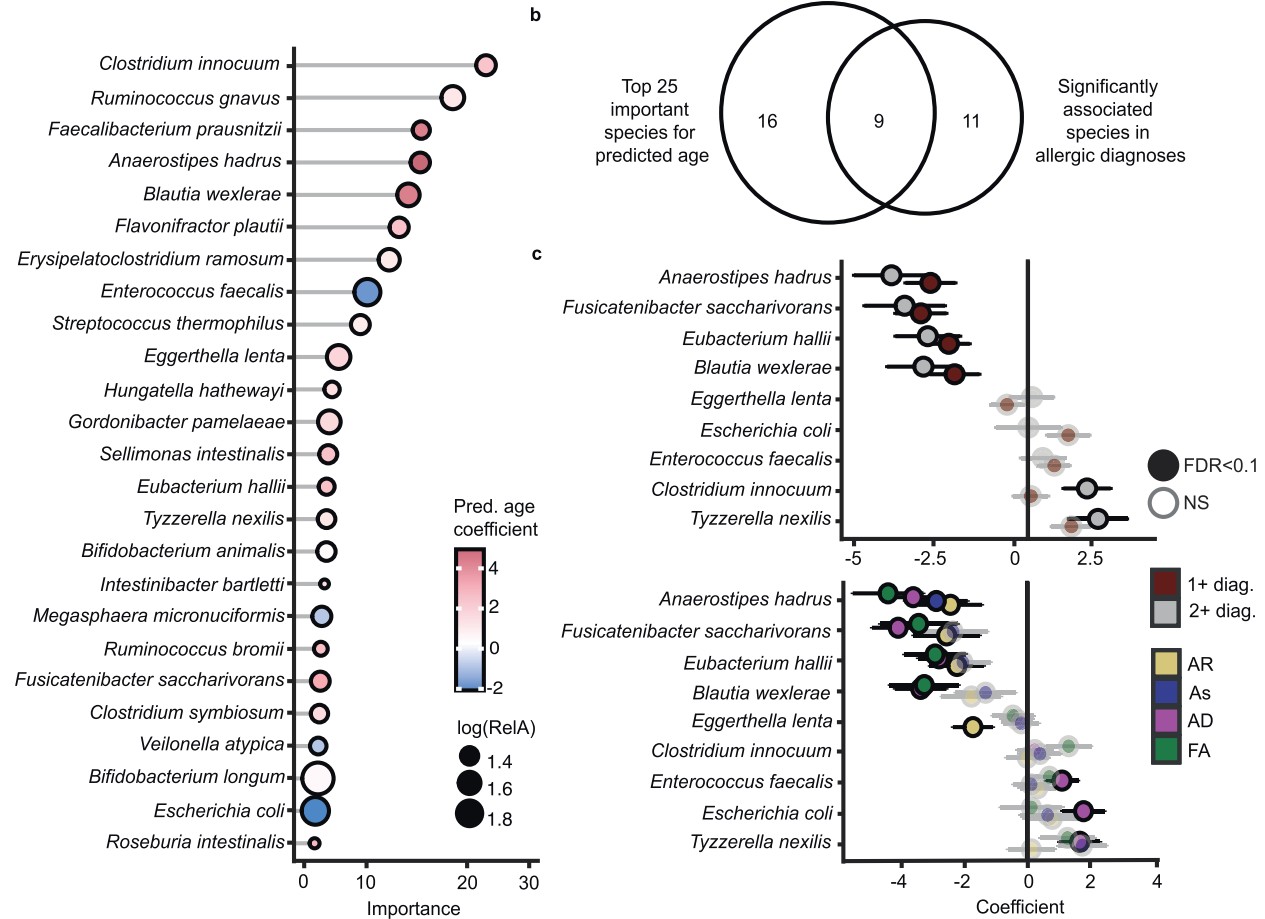

**Fig. 4 | Important underlying microbiota of early-life microbiome age. a** Top 25 most important species in predicted age determination, shaded according to the MaAsLin2 regression coefficient with predicted age, adjusting chronological age and with a random effect of the sample collection site. The size of the points represents logarithmic relative abundance. **b** Venn diagram of the top 25 important species in predicted age and differential within atopic disease as compared to healthy controls. **c** The nine commonly identified species within one or more atopic or allergic diagnoses (1+, $n = 353$), two or more allergic diagnoses (2+, $n = 82$), and individual clinical diagnoses at 5 years, i.e., atopic dermatitis (AD, $n = 212$), food allergy (FA, $n = 75$), asthma (As, $n = 103$), or allergic rhinitis (AR, $n = 113$), at 5 years, as compared to healthy control (HC, $n = 236$) participants, adjusting for chronological age at the time of collection and with a random effect of the sample collection site. Data were presented as MaAslin2 coefficients ± standard error.

whether microbiota maturation is associated with the infant gut metabolome and subsequent allergy development. We applied targeted nuclear magnetic resonance (NMR; 31 metabolites) and liquid chromatography with tandem mass spectrometry (LC-MS/MS; 214 metabolites) to a subset of CHILD participant stool samples ($n = 509$) to address this important aspect of the contributions of the microbiome to the gut environment through the measurement of 245 relevant metabolites (Fig. 1). To initially understand the relationship between microbiota maturation and metabolic profiles at 1 year, we performed a PERMANOVA analysis to quantify the percent of variance explained by microbiota-derived predicted age and adjusting for a time between sample collection and storage and exact age of sample collection, while using collection site as a stratum. We found that microbiota-predicted age significantly explained the variance of the 1-year metabolome (2.2% variance explained, $p = 0.00090$, $F = 15.35$) (Fig. 6a). We next clustered related metabolites using weighted correlation network analysis (WGCNA) into 14 different modules (with 50 metabolites not conforming to any of the 14 modules), and we mapped their interaction to visualize the metabolic landscape of the infant gut at 1 year (Fig. 6b and Supplementary Fig. 5). We identified seven modules and 27 unclustered metabolites, representing 120 of the total 245 metabolites, that were significantly associated with predicted age (FDR <0.1, Supplementary Fig. 5, revealing that, in addition to the functional potential of the

microbiome, microbiota maturation is also strongly related to the infant gut metabolome.

We then assessed the relationship between these metabolite concentrations with both the species and pathways of interest we had identified through metagenomics (Fig. 6c). Amongst the correlations, we identified ten individual metabolites and four metabolic clusters that were significantly related to important pathways within the infant gut microbiota (FDR <0.05). We furthermore identified 11 individual metabolites and seven metabolic clusters that were significantly associated with important species abundance. Of the important functional pathways, sulfur oxidation correlated with the greatest number of metabolic features (six individual metabolites and two metabolic clusters), while of the important microbiota, *E. coli* correlated with the greatest number of metabolic features (five individual metabolites and two metabolic clusters). Of the metabolites, trace amines (TAs) derived from aromatic amino acids, tryptamine, tyramine, and phenylethylamine, significantly correlated with 85% (17 of 20) of the important features, and butyrate, a key short-chain fatty acid in immune tolerance, correlated with *F. saccharivorans*, *A. hadrus*, and sulfur oxidation.

Delayed predicted age is significantly associated with each 5-year allergy diagnosis as well as dysregulation within both microbiome functional capacity (e.g., altered mucous degradation, increased oxidative respiration, and oxidized monosaccharide utilization, as well as

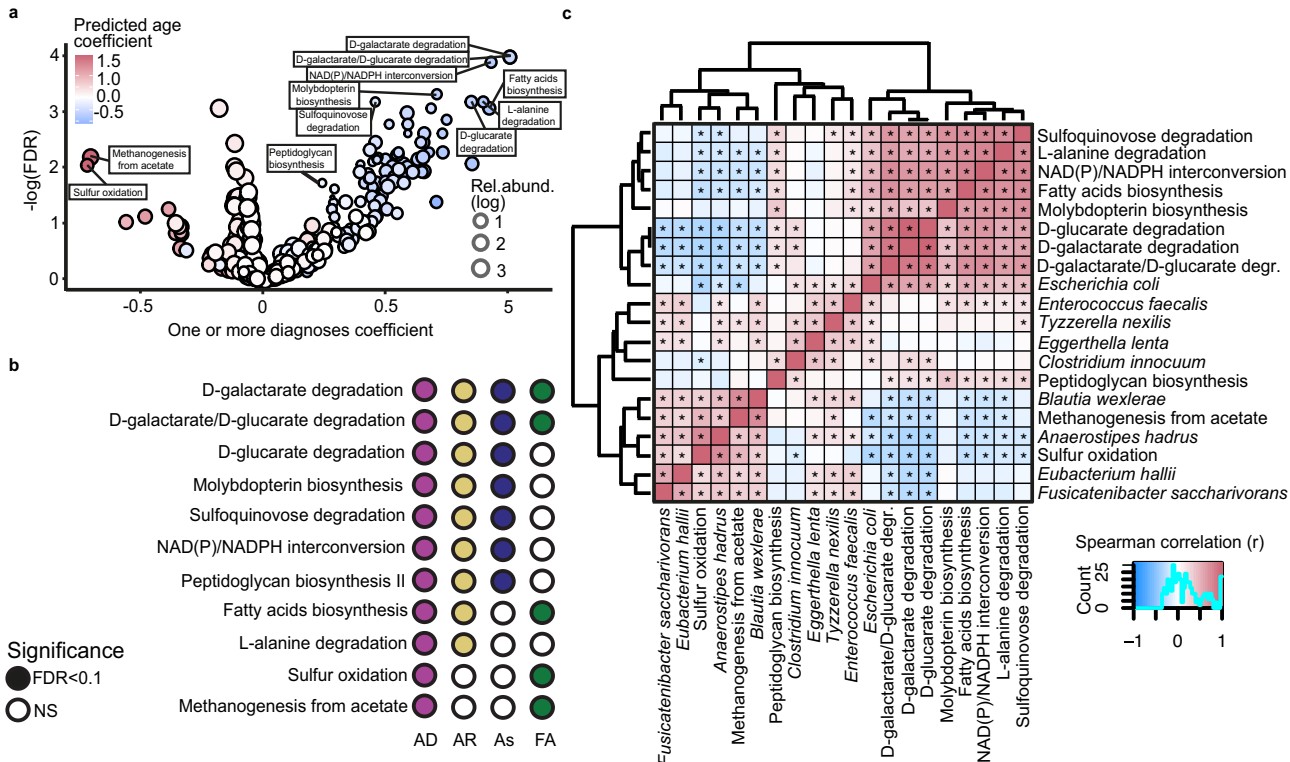

**Fig. 5 | Functional differences within the early-life gut microbiome of infants later diagnosed with atopic diseases. a** MaAslin2 volcano plot of Metacyc-annotated gene pathways using predicted age as the outcome, adjusting for chronological age and a random effect of the sample collection site. **b** 11 pathways associated with predicted age and differential within four or more atopic disease differential analyses (individual or aggregate; FDR < 0.1). **c** Heatmap of spearman correlation analysis between 11 pathways of interest and nine species identified in Fig. 4.

diminished sulfur oxidation and secondary fermentation capacity) (Fig. 5) and the metabolic landscape of the 1-year infant gut (e.g., elevated TAs and decreased butyrate) (Fig. 6 and Supplementary Fig. 5). We, therefore, developed a structural equation model (SEM) to test the hypothesis that this dysregulation in both the genetic potential and metabolic output at 1 year mediates the elevated risk of allergy in children with delayed microbiota maturation (Fig. 7). Dysregulated pathways and metabolites in the 1-year stool sample were combined into one latent variable and an indirect effect was quantified between the microbiota-predicted age and the diagnosis of allergies at 5 years. Within this model, we identified a significant indirect effect ($p = 0.0020$, $\beta = -2.28$) for the 1-year stool latent variable, with each dysregulated feature contributing to this effect. Thus, the association between impaired microbiota maturation and allergies at 5 years is likely mediated by these multi-omic signals, placing them at the forefront of mechanistic targets with the potential to collectively predict and/or prevent allergy development.

## Discussion

Despite having unique organ-specific clinical manifestations, the interrelationship between allergic diseases indicates that common pathophysiological mechanisms contribute to their development[11–15]. Indeed, while many studies focused on individual allergic disorders have identified associated shifts within the microbiota, we found only one other study that took an aggregated approach[16], and no published studies that also investigated microbiota signatures existing prior to allergic sensitization. By combining extensive, longitudinal clinical phenotyping with expert physician clinical assessments throughout the first 5 years of life, we were able to identify the infant microbiome shifts that existed before allergic diagnoses. Furthermore, this approach provided us with a richly characterized "healthy control" group that lacked any detectable signs of allergic sensitization

measured on three separate visits from ages 1 to 5 years. In doing so, we were able to demonstrate that, regardless of the diagnosis, reduced microbiota-predicted age is a hallmark of future allergy development, thus creating a focal point to combat pediatric allergic disease.

While previous studies have associated decreased microbiota maturation with early atopic sensitization and asthma development[23,34,35], our study suggests that this impaired maturation may be universal to the full spectrum of pediatric allergic diagnoses. Our reported trend in maturation alteration is epitomized in depletions in the bacterial species *A. hadrus*, *F. saccharivorans*, *E. hallii*, and *B. wexlerae* in participants who later developed allergic diseases, as well as enrichments in *E. lenta*, *C. innocuum*, *E. faecalis*, *E. coli*, and *T. nexilis* in these participants. The depleted bacterial populations are known short-chain fatty acid (SCFA) producers, notably the butyrate producers *A. hadrus*, *E. hallii*, and *F. saccharivorans*[36–38] and the acetate producer *B. wexlerae*[39]; SCFAs are metabolites that mediate well-defined host benefits within the gut[40]. Although not significantly associated with allergic disease by itself, we report a depletion of butyrate in allergy-prone participants and significant associations between *A. hadrus* and *F. saccharivorans* respective relative abundance and butyrate concentration. This strengthens the postulation that the production of butyrate and its effect on immune cells is a mode by which optimal immune modulation occurs during early life. In contrast, species enriched in allergy-prone participants have been linked to pathogenic activity and poor health outcomes[41–45], with many of these microbiome features associating with metabolites enriched within these same participants.

Furthermore, our mechanistic characterization extended previous knowledge to pinpoint maturation-dependent functional features. We report broad alterations of functional potential for both decreased microbiota-predicted age and the independent considerations of allergic diagnoses. Within the microbiome, imbalances in

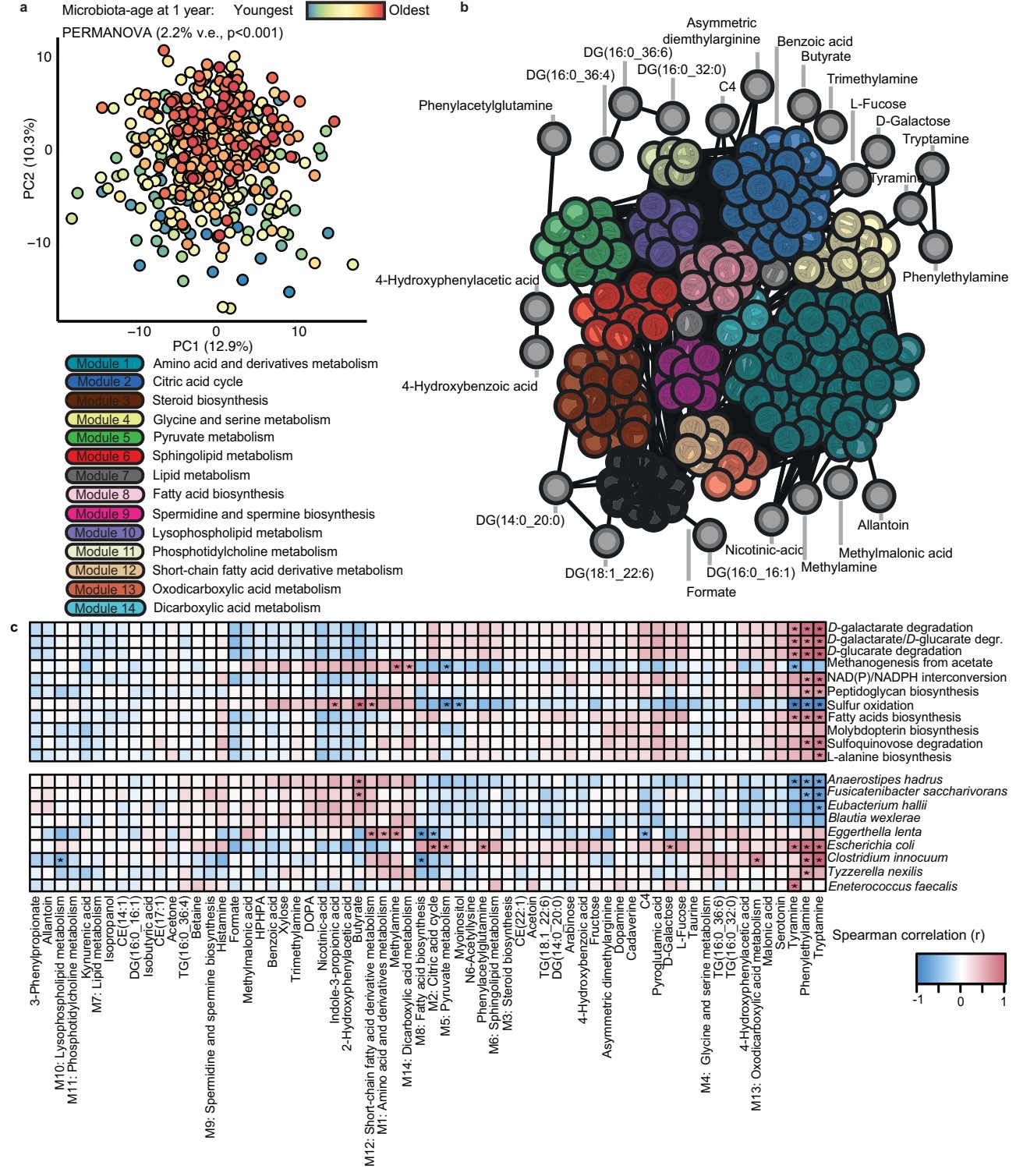

**Fig. 6 | Relating significant microbiome features with metabolic profiles in the gut. a** Principal component analysis (PCA) plot of variance within the 1-year gut metabolome and colored by predicted age distribution. **b** Weighted gene coexpression analysis (WGCNA)-determined modules and interactions of metabolites in the 1-year gut, mapped using Cytoscape. **c** Spearman correlation heatmap of the relationship between metabolites, WGCNA clusters, and microbiome features of interest identified in Figs. 3, 4. (*) $q < 0.05$.

functional pathways indicative of dysbiosis were associated with reduced predicted age and elevated risk of allergy. These included a breakdown of mucous integrity via elevated sulfur reduction and diminished sulfur oxidation pathways[46], elevated oxidation levels and subsequent availability of oxidized monosaccharides[27,47], as well as reduced potential for secondary fermentation[48]. Our shotgun

metagenomic sequencing results were complemented by targeted metabolic profiling within the same infant stool samples. In addition to a high degree of connectivity between microbiota maturation and the functionality of the microbiome, we discovered a strong association between microbiota-derived age and the stool metabolic landscape. Indeed, of the 245 metabolites studied, 27 of the 50 (54%) cluster-

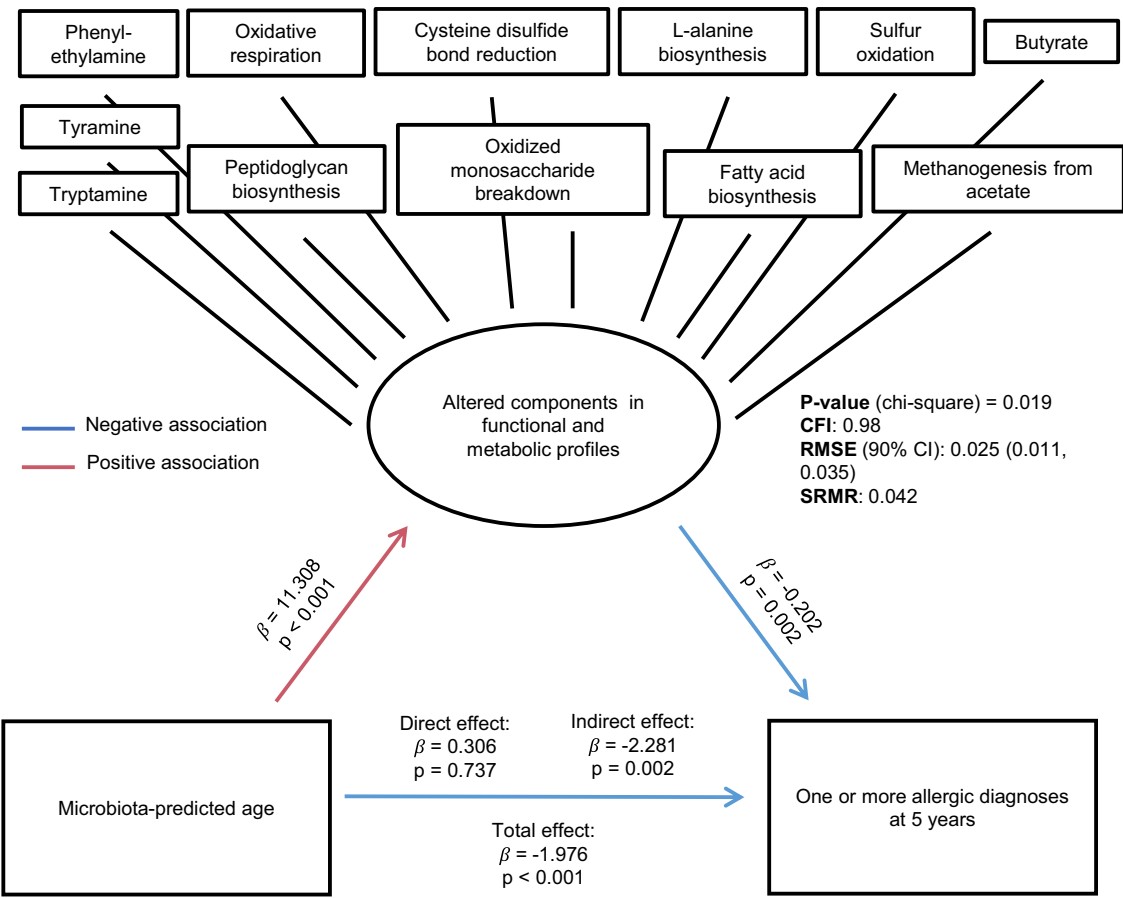

**Fig. 7 | Linking predicted age and allergic disease using specific microbial and metabolomic features.** Structural equation modeling (SEM) diagram showing the direct and indirect effects of predicted age upon atopic and allergic disease, as mediated by the 1-year microbiome and metabolome features.

independent metabolites and 7 of the 14 (50%) metabolite clusters were significantly associated with the microbiota-predicted age. Several maturation-dependent metabolites were disturbed in children who developed allergies with three biogenic amines, namely phenylethylamine, tryptamine, and tyramine, also demonstrating high correlations with disrupted microbial pathways. Typically found at low levels, trace amines (TAs) phenylethylamine, tryptamine, and tyramine have distinct biological impacts compared to other biogenic amines. Indeed, TAs have a very high affinity for TA-associated receptors (TAARs), a class of G-coupled protein receptors found on both intestinal and immune cells, and TAAR ligation has been demonstrated to increase intestinal cell oxidative stress and immune cell activation[49,50]. Furthermore, their accumulation promotes bacterial adherence to intestinal cells, likely perpetuating this inflammatory response[51,52]. In this way, microbiota-dependent limiting of TA abundance may be an underappreciated mechanism to promote tolerogenic immune development in infancy.

Within this study, we explicitly acknowledge that we are reporting epidemiological associations and that our stringently selected cohort ($n = 1115$) does not completely represent the entire CHILD cohort (e.g., we report a slightly higher proportion of participants with familial history of atopy and history of breastfeeding than the larger cohort) (Supplementary Table 1). Moreover, CHILD is a prospective observational cohort, and additional studies from independent cohorts are needed to strengthen our findings, as is the case for most microbiome-based studies. This is primarily due to natural variance that exists within and between samples. For example, biogeography within stool samples can impact microbiota composition, and while stool aliquots were homogenized prior to our metagenomic and metabolomic

analyses, variation may exist between individual aliquots collected from the same stool. Variance is also reflected between samples (e.g., infant stool may differ in composition depending on the day, and infants across different populations often demonstrate distinct microbiota composition). Thus, while our findings are promising and were observed in a large, cross-Canada cohort, replication within other well-powered cohorts will be important for validating these results.

Although we report a potential mechanism for microbiota-dependent support of tolerogenic immune development via the interaction of the elevated TAs tryptamine, tyramine, and phenylethylamine, and depleted butyrate with intestinal and immune cell receptors in infants later diagnosed with allergic disease, future studies are needed to perform mechanistic in vitro and in vivo studies in cell and murine models to provide insight into causation. This is especially applicable to the presence of metabolites within participant stool samples and their attribution to specific microbial species and/or functional pathways, as we were only able to identify correlations between metabolites and pathways. Mechanistic studies capable of testing species-specific enzymatic relationships with these metabolites would greatly improve our understanding of their relationship with microbiota maturation and allergic disease.

Although we emphasized the similar clinical and microbiome signals underlying allergic disease, identifying unique features associated with individual diseases and atopic and non-atopic manifestations provides valuable insight into differentiating their biological underpinnings, and further work in this area of study is needed. Studies looking more closely at clinical and environmental variables that are associated with microbiota maturation and allergy development would therefore be beneficial.

Overall, using deep clinical phenotyping, including repeated testing and standardized physician diagnoses, we compared 1115 children with asthma, allergic rhinitis, food allergy, or atopic dermatitis to a rigorously defined, non-allergic comparator group. We then harnessed a multi-omics approach in 589 of these children containing shotgun metagenomic sequencing and their infant microbiome metabolomic profiles and revealed that impaired infant microbiota maturation may be universal to the development of all pediatric allergies. We described detailed underpinnings driving this decrease in gut microbiome maturation, encompassed within the alteration of a core group of species, functional pathways (i.e., potential intestinal mucous integrity breakdown, elevated oxidative stress levels, and subsequently oxidized monosaccharides, and diminished secondary fermentation), and metabolic imbalance (i.e., elevated TAs) associated with reduced microbiota-maturation age and elevated risk of allergy. In conclusion, this study provides insight into underappreciated and nuanced aspects of the infant microbiome that will enable improved prevention and prediction of allergic disease.

## Methods

### Study cohort and defining clinical phenotypes

This research complies with all relevant ethical regulations and was written and approved by the University of British Columbia, University of Manitoba, University of Toronto, McMaster University, BC Children's Hospital, The Hospital for Sick Children, and Simon Fraser University. The Research Ethics Board Number is H07-03120. The Board of Record (as noted above) has reviewed and approved this study in accordance with the requirements of the Tri-Council Policy Statement: Ethical Conduct for Research Involving Humans (TCPS2, 2018). The "Board of Record" is the Research Ethics Board delegated by the participating REBs involved in a harmonized study to facilitate the ethics review and approval process.

The CHILD Study is a multi-center longitudinal, prospective, general population birth cohort study following infants from pregnancy to age 5 years, and beyond. With enrollment beginning in 2008 and closing in 2012, a total of 3621 pregnant women from four cities (Vancouver, Edmonton, Winnipeg, Toronto) across Canada enrolled along with eligible infants ($n = 3455$) that had no congenital abnormalities and were born at a minimum of 34 weeks of gestation[53]. Informed consent was obtained from parents at the time of this study. CHILD Study children were followed prospectively and detailed information on environmental exposures and clinical measurements and assessments were collected using a combination of questionnaires and in-person clinical assessments. Briefly, questionnaires were administered at recruitment, 36-week gestation, at 3, 6, 12, 18, 24, 30 months, and at 3, 4, and 5 years; data were obtained related to environmental exposures and general health. In addition, at ages 1, 3, and 5 years, questionnaires validated in the International Study of Asthma and Allergies in Childhood (ISAAC)[54] were completed by the parent.

All infants enrolled in the CHILD protocol were administered an SPT at their 1-, 3-, and 5-year scheduled visits. Children were then diagnosed with IgE-mediated allergic sensitization (also referred to as atopy) based on skin prick testing (SPT) to multiple common foods and environmental inhalant allergens, using ≥2 mm average wheal size as indicating a positive test relative to the negative control. The allergens tested at all 1-, 3-, and 5-year visits include cat hair, the German cockroach, Alternaria tenuis, house dust mites (*Dermatophagoides psteronyssinus* and *Dermatophagoides farnae*), dog epithelium, cow's milk, peanut, egg white, and soybean. In addition to these, participants at 3- and 5-year visits were tested with *Cladosporium*, *Penicillium*, *Aspergillus fumigatus*, trees, grass, weeds, and ragweed. Glycerin and histamine served as the negative and positive controls, respectively.

The primary outcomes of our study were atopic dermatitis, asthma, food allergy, and allergic rhinitis diagnosed (as Yes/Possible/No), using history and physical examination in combination with skin prick testing, by an expert study physician at the clinical assessment at the age of 5 years based on our published approach[55]. For this study, children were considered to have allergic diseases only if the response was 'Yes' to a clinical inquiry of whether they had atopic dermatitis, asthma, food allergy, and/or allergic rhinitis. Non-allergic controls were limited to children with 'No' responses for 5-year diagnoses, negative allergen SPTs at 1, 3, and 5 years, and no history of wheezing at 1, 3, and 5 years. Within the current study, we analyzed the data in a subset of CHILD that contained data for parent and child questionnaires, SPT results, and physician diagnoses at 5 years for cases and 1, 3, and 5 years for controls ($n = 1115$, Fig. 1b and Supplementary Table 1).

### Stool sample collection

Sample collection and sequencing were performed as previously described in refs. 19,53. Specifically, stool samples from diapers were collected at a home visit at around 3 months [mean (SD), 3.8 (1.1) months] and a clinic visit at around 1 year [mean (SD), 12.5 (1.6) months]. Samples were briefly stored at 4 °C and then aliquoted into four 2-mL cryovials using a stainless steel depyrogenated spatula and were frozen at −80 °C. CHILD recorded the time between stool collection and long-term storage, and this processing time was adjusted for in our statistical analysis of metabolic profiles.

### Shotgun metagenomic sequencing

Shotgun metagenomic sequencing data were generated by Diversigen (Minneapolis, MN, USA) from fecal samples (average depth of 5 million reads per sample). DNA was extracted from samples using the MO BIO PowerSoil Pro with bead beating in 0.1 mm glass bead plates, with high-quality input DNA verified using Quant-iT PicoGreen. Libraries were prepared and sequenced on an Illumina NextSeq using single-end $1 \times 150$ reads. Low-quality ($Q$-score <30) and length (<50) sequences were removed, and adapter sequences were trimmed. Host and low-quality reads were removed, and only samples with a minimum of 1 million remaining reads were retained for downstream analysis.

**Sequencing preprocessing.** The bioBakery 3 pipeline was used to map sequences and classify sequences into taxonomic (species and strain level) and functional features within each sample[56]. The bioBakery 3 pipeline is open source and its functionality has been published[56]. Specifically, MetaPhlAn 3 was used for taxonomic classification, and HUMAnN 3 for functional profiling.

### NMR and LC-MS/MS metabolite quantification

Metabolic profiles were created from the same sequenced stool samples at The Metabolomics Innovation Center (TMIC) in Edmonton, Alberta using two separate assays. Targeted nuclear magnetic resonance (NMR) analysis of 31 metabolites was performed across 62 batches[57,58]. Targeted liquid chromatography with tandem mass spectrometry (LC-MS/MS) analysis of 590 metabolites was performed using TMIC's Microbiome Metabolism (MEGA) assay across 27 batches[59,60]. NMR and LC-MS/MS precision were confirmed to be <5 and <10% coefficients of variability (CV), respectively. Additionally, overlapping metabolites detected by both methods were cross-checked to confirm the accuracies of the reported concentration values.

Detailed methods of both NMR and LC-MS/MS analyses can be found in Supplemental Methods. Briefly, each analysis was performed using approximately 100 mg of stool. Samples with low mass or diaper fibers were excluded. Stool was weighed before and after lyophilization to quantify total water content prior to analysis. Analyte concentrations were determined using a standard approach to absolute quantification, using isotope-labeled internal standards to correct for technical variation and then assessing the result against a calibration curve of known concentrations of standard mixtures. Regarding the

assignment of all visible peaks, this applies to when, within this targeted assay, very low abundant peaks are not visible for manual assignment and are therefore not reported Metabolite levels (μmol) were normalized to dry/lyophilized stool weight (g) and analyzed using the ratio (μmol/g). All metabolite concentrations for both the NMR and LC-MS/MS analysis were recorded by TMIC as well as their limit of detection (LOD).

All 31 metabolites from the NMR analysis were kept for downstream analysis. Of the 590 metabolites targeted in the LC-MS/MS analysis, we excluded metabolites that were detected below the limit of detection in more than 80% of samples (meaning they were present in less than 20% of our samples). This resulted in the exclusion of 244 metabolites. Remaining metabolite concentrations below the limit of detection (LOD) were imputed with a value of one-half the minimum concentration for each metabolite and log-transformed. An additional 132 low-variance metabolites based on standard deviation (log(SD) less than −5) were then excluded. All excluded metabolites were confirmed to not be significantly associated with the presence of a 5-year allergy diagnosis (Supplementary Data 4).

Technical sample outliers were detected via PCA analysis followed by the quantification of local outlier factor (lof), using the "stats" and "dbscan" packages, respectively. Samples with a lof greater than 5 were excluded (three LC-MS samples and two NMR samples). The resulting NMR and LC-MS/MS were individually batch corrected using the "ComBat" package prior to any downstream analysis. This reduced the effect of the batch from $R^2 = 0.11$ to $R^2 = 0.017$ in the LC-MS/MS dataset and reduced the effect of the batch from $R^2 = 0.13$ to $R^2 = 0.004$) in the NMR dataset. Batch-corrected datasets containing a total of 245 metabolites were merged for downstream analyses. These pre-processing steps have been included in our supplementary code files (R file and README) for clarity of data treatment and replication.

## Statistics and reproducibility

Data analysis was conducted in R (version 4.1.1). No statistical method was used to predetermine the sample size. The Investigators were not blinded to allocation during experiments and outcome assessment. Firstly, we derived variables indicating whether participants were diagnosed with a condition of interest or had no conditions up and through their 5-year evaluation and used multivariable conditional logistic regression (stratified by study center) to evaluate the influence of early-life and familial exposures, including biological sex, presence of older siblings, mode of delivery at birth, birth weight, the season of birth, breastfeeding status at the age of 6 months, maternal atopy, paternal atopy, and exposure to environmental $NO_2$, upon allergic condition development by the age of 5 years. Missing data were considered missing completely at random, and individuals were removed from the multivariable analysis if they had a missing value in any covariates.

The microbiome estimated age was derived from species-level microbiome relative abundance data using a nested fivefold cross-validated random forest regressor. Each measurement was taken from a distinct sample for the calculation. By using the "randomForest", "mlbench", and "caret" packages in R[61-63], a fivefold random forest model with an mtree value of 500 was used to predict the exact age using the relative abundance of species with at least 10% prevalence within all samples. Within each fold of the cross-validated analysis, the hyper-parameters of the random forest model were tuned given a grid search space using nested fivefold cross-validation. The predicted microbiome age was the combination of the predicted value of each holdout set from each cross-validated random forest regression. The importance of features was the average of importance from all models in the nested cross-validated regressor.

The "phyloseq" package was used to pre-process the metagenomic taxonomy table[64]. The "Maaslin2" package was used to perform linear mixed-effects models (MaAsLin2 function)[65] with study center location as a random effect and adjusting for stool sample of collection age to examine the association between microbial community structure at 3 months and 1 year of age within the six phenotypes as compared to the control group of participants. These phenotypes included having at least one of atopic dermatitis, asthma, food allergy, or allergic rhinitis, having at least two of atopic dermatitis, asthma, food allergy, or allergic rhinitis, and having atopic dermatitis, asthma, food allergy, or allergic rhinitis individually. Specifically, comparisons of each group to participants determined to have no allergic conditions up and through their 5-year evaluation were performed independently of one another.

Species diversity, measured as the Shannon index, was calculated using the "vegan" package. To identify bacterial species and MetaCyc pathways significantly associated with each phenotype, the same model was applied to the log-transformed relative abundance of species and MetaCyc pathways. Models were only applied to species and MetaCyc pathways detected in at least 10% of used samples were tested (72 species and 347 pathways, respectively), the default setting in the MaAsLin2 package. MaAsLin2 adds a pseudocount of half the minimum species or MetaCyc pathway level detected before the log transformation relative abundances. *P* values were corrected using the Benjamini–Hochberg approach and results with FDR <0.1 were considered significant and presented as so.

For all metabolomic profile analyses, batch corrections were performed prior to analyses. Permutational Multivariate Analysis of Variance (PERMANOVA) analysis was applied to quantify the association between infant microbiota-derived predicted age and metabolome using the R package "vegan"[66]. Euclidean distance was used as the metric of comparison between the predicted age with adjustment for the exact age of sample collection and processing time between stool collection and long-term storage, and collection site as a stratum. Spearman correlation analyses were performed using the "RcmdrMisc" package and reported using *r* and Benjamini–Hochberg-corrected *p* values[67].

Meconium metabolite abundances were log-transformed and then weighted gene coexpression network analysis (WGCNA) was performed using the "WGCNA" package in R[68]. A soft-power threshold of 4 was selected based on the scale-free topology fit index. Positively correlated metabolites were clustered together using "signed hybrid" networks and biweight midcorrelation. The minimum module size was set to five metabolites, and modules that correlated with each other at 0.85 or greater were merged. This resulted in 14 modules. Scaled average expression values from the WGCNA output were combined with scaled abundances of the 50 unclustered metabolites.

To evaluate the mediation effect of dysregulated pathways and metabolites for gut maturity on allergic diseases, we applied structural equation modeling (SEM) using the R package "lavaan"[69]. For model specification, first, we conceptualized the 1-year gut microbiome imbalanced pathways and metabolites (i.e., latent variable) using factor analysis. The latent variable was defined as a combination of pathways significantly associated with at least two of the allergy diagnoses and one of the composite groups and metabolites associated with a high proportion of those same pathways. Then, we simultaneously estimated the mediation effect (indirect effect) of dysregulated pathways and metabolites by fitting two multiple regressions on latent and outcome variables, assuming that both regressions' error terms are uncorrelated. All the models were adjusted for the study center, stool sample collection age, and processing time between stool collection and long-term storage.

## Reporting summary

Further information on research design is available in the Nature Portfolio Reporting Summary linked to this article.

## Data availability

The participant data were available under restricted access for the protection of CHILD participants, access can be obtained by contacting Stuart E. Turvey (sturvey@bcchr.ca). The shotgun metagenomic data used in this study are available in the NCBI database under BioProject accession code PRJNA838575. The metabolic profile data used in this study are available in the MetaboLights database under accession code MTBLS7919. For additional participant clinical and stool data, further requests for resources and reagents should be directed to and will be fulfilled by Stuart E. Turvey (sturvey@bcchr.ca).

## Code availability

The code for the study is provided in the Supplementary Files.

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

## Acknowledgements

We are grateful to all the families who participated in this study and the CHILD team, including interviewers, nurses, computer and laboratory technicians, clerical workers, research scientists, volunteers, managers, and receptionists. The Canadian Institutes of Health Research (CIHR), Debbie and Don Morrison and the Allergy, Genes, and Environment Network of Centres of Excellence (AllerGen NCE) provided core support to establish the CHILD Study (grants to founding CHILD director MS and current director PS). S.E.T. holds a Tier 1 Canada Research Chair in Pediatric Precision Health and the Aubrey J. Tingle Professor of Pediatric Immunology. C.H. is funded by the University of British Columbia John Richard Turner Fellowship in Microbiology, the President's Academic Excellence Initiative PhD Award, and the University of British Columbia Four Year Doctoral Fellowship (4YF). D.L.Y.D. is funded by the Canadian Institute of Health Research Frederick Banting and Charles Best Canada Graduate Scholarship Doctoral Award (CIHR CGS-D) and the University of British Columbia 4YF. M.B.A. holds a Tier 2 Canada Research Chair in the Developmental Origins of Chronic Disease and is a Fellow of the CIFAR Humans and the Microbiome Program. Additional funding was provided by CIHR, AllerGen NCE, Genome Canada, and Genome British Columbia ([274CHI]) S.E.T., [FDN-159935] B.B.F., and [EC1-144621] S.E.T.). P.S. holds a Tier 1 Canada Research Chair in Pediatric Asthma and Lung Health. We further acknowledge the support of BC Children's Hospital Research Institute and Foundation, and the Provincial Health Services Authority.

## Author contributions

Conceptualization, C.H, D.L.Y.D., C.P., and S.E.T.; Methodology, C.H, D.L.Y.D., and C.P.; Investigation and formal epidemiological, metage-nomic, and metabolomic analysis, C.H., D.L.Y.D., and C.P.; Visualization, C.H. and C.P.; Funding acquisition, S.E.T.; Resources, M.B.A., B.B.F., P.S., and S.E.T. Data oversight—Statistical analyses, C.H., D.L.Y.D., and C.P.; Unrestricted access to all data, C.H., D.L.Y.D., C.P., and S.E.T.; First draft, C.H., D.L.Y.D., C.P., and S.E.T.; Review and editing, C.H., D.L.Y.D., A.B.B., T.J.M., P.J.M., B.B.F., E.S., A.L.K., P.S., M.B.A., C.P., and S.E.T. All authors agreed to submit the manuscript, read and approved the final draft and take full responsibility of its content, including the accuracy of the data and its statistical analysis.

## Competing interests

The authors declare no competing interests.
