## [Peer Review File · Nature Communications]

REVIEWER COMMENTS

Reviewer #1 (Remarks to the Author):

The authors longitudinally investigated the gut microbiome metagenomic and metabolomic profiles in children diagnosed with 4 distinct allergies (atopic dermatitis, asthma, food allergy and allergic rhinitis) and in healthy children. They considered a total of 1115 samples, as part of the CHILD study cohort, which includes extensive metadata for the first 5 years of life. The authors found that impaired gut microbiome development at 1 year of age was associated with the development of a multiple allergies in scholar age (5yrs). The dysregulation in terms of microbial composition was reflected also in the microbiome functional capacity. Overall, the impaired multi-omic signatures were ubiquitous to all allergic diseases investigated in this study.

Major comments:

There are no major comments. The study is a relevant addition to the field, the study design is solid and the analysis is well executed.

Minor comments:

- The authors should provide a study design overview figure, to help the reader quickly understand i) how many samples from each condition and from healthy children were included in the analysis, ii) the sampling timepoints (3 months, 1 year, 5 years?) and how many samples per timepoint, and iii) which analysis (metaG and/or metaB) was performed on how many samples at each timepoint.
- The authors should consider expanding their comments on the species that were found to be decreased or increased in infants that got later diagnosed with allergic disease. *A. hadrus* and *E. hallii* are both known to be important SCFA producers, while *E. coli*, *E. faecalis* and *C. innocuum* have the potential to be pathogenic (or at least some strains do).
- Figure 1A: this panel should be a table instead, with the plot showing only the adjusted odds ratio and associated p-values.
- Figure 5C: upper and lower panel are not aligned

Reviewer #2 (Remarks to the Author):

The authors report comprehensive gut microbiome data from a well-characterized birth cohort, i.e. the CHILd cohort in Canada (n=1,115). Four allergic diagnoses at 5 years: atopic dermatitis (AD, n=367), asthma (As, n=165), food allergy (FA, n=136), and allergic rhinitis (AR, n=187) were assessed by study physicians or highly trained health care professionals under their supervision. In a subset of about half the population (n=589) shotgun metagenomic and metabolomics profiling of fecal samples at ages 3 and 12 months was performed. Impaired 1-year microbiota maturation was related to risk of all allergic outcomes and a core set of functional and metabolic features was identified mediating the effect.

Comments:

1. There is a long-standing debate whether all so called atopic diseases which have very distinct clinical features (skin or upper and/or lower respiratory tract involvement with and without concomitant allergic sensitization to food and/or aeroallergens) can be lumped together or whether they show divergent underlying pathomechanisms. One feature that may be common to many but not all of them is allergic sensitization which was assessed with a low cut-off (skin prick test ≥ 2 mm after subtraction of the negative control). Given that in the overall cohort “The majority of parents reported a history of ever having allergies and atopic diseases (77.0%)...” (Reference 45) such enrichment for a family history of allergies may have contributed to a high proportion of allergic sensitization in this population which may be the common feature of all assessed allergic outcomes. The authors should therefore show in the Venn diagram in supplemental figure 1 the additional overlap with allergic sensitization. If there is a strong overlap, then it will be very hard to disentangle the single clinical disease entities from allergic sensitization and the true association with delayed microbiota maturation may be with allergic sensitization rather than all the mentioned diseases. Given the smaller sample size (n=589) with the MetaCyc and metabolomics data, the number of subjects with clinical diagnoses without allergic sensitization will likely be too small to allow robust statistical analyses. Alternatively, stratification of analyses by allergic sensitization at any of the three assessed time points may help, but may suffer from the same limitation.
2. Any underlying other feature common to all allergic outcomes such as a family history, breastfeeding and antibiotic usage as shown in figure 1 A/B or persistent sensitization as defined in the LCA in reference 45 rather than any sensitization may be the true factor associated with microbiome maturation. In this case, mediation analyses between the potential underlying tertium comparationis and the outcomes would be more appropriate.
3. Further along this line of reasoning: T which life style factors (family history, breastfeeding, antibiotic use) determined the 9 taxa associated with allergy development and protection, respectively?
4. The CHILd cohort followed 3495 enrolled subjects. The authors must report if any selection bias, e.g. further enrichment of a positive family history or of allergic sensitization/outcomes occurred in the subsample of n=589.

5. The methods section does not report allergic sensitization to pollen, yet in reference 45 SPTs to also aeroallergens are mentioned for age 3 years. How many children were ONLY sensitized to tree and/or grass but not to other allergens? In other words, how many children would have been non-atopic at age 3 years when excluding the children ONLY sensitized to tree and/or grass?
6. The authors report a decrease in Shannon diversity at age 3 and age 12 month – was such association also seen for other measures of alpha diversity?
7. Most of the manuscript focuses on the maturation at age 12 months. The investigators however also collected fecal samples at age 3 month. Was there any association between microbiome features, in particular alpha-diversity at age 3 month at the maturation as assessed by age 12 month?
8. According to figure 5c tryptamine measured in fecal samples was strongly positively correlated to most of the 11 Metacyc-annotated gene pathways. Yet, in figure 6 they appear as independent contributors in the structural equation modeling diagram. How can individual independent effects be disentangled for each individual contributing factor? This also applies to the discussion section where much emphasis is put on the trace amines.
9. Line 185: I assume this relates to 12 mo fecal samples?

Reviewer #3 (Remarks to the Author):

Hoskinson et al have studied the differences in gut microbiota between children with allergic disease and compared them to a control group. The study is interesting and creative in the way it has addressed the problem and I think will be of interest to Nature Comms readers. I very much enjoyed reading it.

I have concentrated on the metabolomics analysis as my area of expertise. The reasoning of the paper is clear, although more details on how the technical aspects of metabolomics may be influencing the results should be considered. The conclusions have been drawn on a single 2 mL aliquot of stool. This is common practice in this field, but there is good evidence to suggest that certain metabolites may not be homogeneously spread throughout the stool. In addition, the methods for stool preparation are not well described, but according to their cited paper Moraes et al, may have been refrigerated for up to three days. This will certainly affect the metabolome, and may also have affected the microbiome. Was any attempt made to assess and control for length of refrigerated storage? What data have you to show that your method gives reliable, repeatable results?

The metabolomics protocol does not contain enough information to allow the experiment to be repeated. Specifically, it does not give any information on extraction methods. There should be more details provided in the supplementary on the extraction techniques and the metabolites which were detected by NMR and MS and whether any overlapped. The results cannot be reliably assessed

without more information on number of batches, inter and intra batch effects and assessment of technical biases in the datasets.

Supp methods section:

For Stool preparation methods, please include the following details:

Number of batches run for each method

Batch differences – how they were assessed and corrected

Carryover or background levels of metabolites – how were they assessed and corrected for.

Number of metabolites targeted with each method

Number found for each method

Number which were measured in the quantitative range for each method.

LC-MS analysis supp method

Red PEEK tubing – this is not very informative since different companies use different colours to denote different internal diameters. Please include company and internal diameter and length of tubing.

LC-MS analysis – please give more details about the calibration concentration ranges, number of calibration points and whether standards were run as individual standards or as mixed standard calibrants. What was the matrix that the calibrants were analysed in and how was background and carryover checked and accounted for. How was identification carried out?

DFI – unexplained acronym

NMR analysis supp method

“Typically all of visible peaks were assigned” – this is described in the methods as a targeted method, but this appears to be untargeted. Please clarify. Also give an indication of background subtraction methods if used, how many peaks were discovered in the average sample and what level of matching similarity was required for identification.

Main paper:

1) How was this selection of stool samples chosen: randomly, a select group or was there any known bias in the selection.

2) How much stool was used, and what efforts were made to make sure it was a representative sample, especially in the original collection period.

3) The variance is relatively small, percentage wise. Given the inherent difficulties with achieving representative faecal samples, what measures were taken to validate this result with an independent cohort.

Line 247: you make a correlation between number of significant metabolites found and relative importance of the pathways they map to. However, whether metabolite concentrations change can also be a function of rate limiting steps in the metabolic pathway. This is more difficult to measure in a mixed microbial environment, but was it considered when analysing the results?

Line 411 unclear – “water content stool weight” – do you mean wet weight of stools?

Has the method been specifically validated for stool, and are the concentrations accurate, especially if the standards are run in a matrix free solvent.

413: missing values can bias a dataset: how many missing values were there as a percentage of the dataset, and were they missing at random or missing not at random.

I think you have used a standard approach to absolute quantification, using an internal standard to correct for technical variation and then assessing the result against a calibration curve. However, the way it is written here sounds as if you are calculating a ratio and then assessing it against a non-ratio measurement to determine the concentration. Perhaps reconsider the wording.

I think to publish in a journal such as Nature Communications, there is a clear expectation that data should be freely available on a site such as Metabolytes and not require the reader to contact the author. This should preferably include the raw data since this is important to assess batch effects, contaminations, incorrect identifications etc.

Author contributions: it should be more clearly set out who carried out the metabolomics analyses.

POINT-BY-POINT RESPONSE TO REVIEWER COMMENTS

Comments from Reviewer 1

Reviewer #1 (Early life microbiome, metagenomics, mother-infant, longitudinal):

The authors longitudinally investigated the gut microbiome metagenomic and metabolomic profiles in children diagnosed with 4 distinct allergies (atopic dermatitis, asthma, food allergy and allergic rhinitis) and in healthy children. They considered a total of 1115 samples, as part of the CHILD study cohort, which includes extensive metadata for the first 5 years of life. The authors found that impaired gut microbiome development at 1 year of age was associated with the development of a multiple allergies in scholar age (5yrs). The dysregulation in terms of microbial composition was reflected also in the microbiome functional capacity. Overall, the impaired multi-omic signatures were ubiquitous to all allergic diseases investigated in this study.

Major comments:

There are no major comments. The study is a relevant addition to the field, the study design is solid and the analysis is well executed.

We thank the reviewer for their kind sentiment.

Minor comments:

- The authors should provide a study design overview figure, to help the reader quickly understand i) how many samples from each condition and from healthy children were included in the analysis, ii) the sampling timepoints (3 months, 1 year, 5 years?) and how many samples per timepoint, and iii) which analysis (metaG and/or metaB) was performed on how many samples at each timepoint.

Thank you for your comment and suggestion. We have taken your advice and amended our original Supplementary Fig. 1 and moved it to the main manuscript as new Figure 1. In this figure we include a timeline of sample and clinical variable collection. We also have clearly defined the participant and sample numbers for our clinical, metagenomics and metabolomics analyses. We additionally added a Venn diagram for each analysis that shows the breakdown of the 4 allergy diagnoses. We hope that this provides sufficient clarification on the study design. The reviewer's comment has significantly improved the study design overview. Please see the new Figure 1 below.

New Fig 1. Clinical evaluation of CHILD participants and data collection from biological samples. **a** Timeline of CHILD enrolment and clinical evaluations from gestation through 5-year evaluations. **b** Consort diagram of CHILD participants and samples included in this study, including the composition of participant allergic diseases and their interrelated diagnoses.

- The authors should consider expanding their comments on the species that were found to be decreased or increased in infants that got later diagnosed with allergic disease. *A. hadrus* and *E. hallii* are both known to be important SCFA producers, while *E. coli*, *E. faecalis* and *C. innocuum* have the potential to be pathogenic (or at least some strains do).

Thank you for this comment. We completely agree that these species deserve more discussion, particularly with regards to SCFA abundance. We had originally had a difficult time pulling out a clear SCFA signal with our metabolomics data. However, thanks in part to the 3rd reviewer’s recommended improvements to how we were addressing technical variation on our metabolomic profiles, we are now able to see a clear connection between butyrate abundance and predicted age. Supporting your comment, butyrate is also significantly associated with *F. saccharivorans*, *A. hadrus*, and sulfur oxidation pathways (Fig. 6c). These results have now been added to the results section of the manuscript within our newly revised Fig. 6 and we also added the additional comments within the Discussion section on **Line 305-316**. Thank you again for the push to dig a little deeper.

- Figure 1A: this panel should be a table instead, with the plot showing only the adjusted odds ratio and associated p-values.

Thank you for this comment. We do prefer the current forest plot that combines numbers, statistics, and a visual representation all in one handy plot. However, we have created a new table as requested. It is currently in the supplemental tables, but we will replace the forest plot with this table if the reviewer/editor feel this will improve the manuscript. Please see the new table below.

Variable	Adjusted odds ratio	Adjusted odds ratio confidence interval	P-value
Sex (male)	1.84	(1.36, 2.49)	6.8E-05
Ethnicity (Non-Caucasian)	2.26	(1.64, 3.1)	5.1E-07
C-section (with labor)	0.71	(0.45, 1.12)	0.14
C-section (without labor)	1.56	(0.96, 2.53)	0.075
Breastfeeding status at 6 months	0.66	(0.45, 0.99)	0.043
Season of birth (Summer)	0.89	(0.59, 1.34)	0.59
Season of birth (Fall)	1.28	(0.83, 1.97)	0.27
Season of birth (Winter)	0.99	(0.65, 1.5)	0.96
Paternal atopy	1.56	(1.13, 2.15)	0.007
Maternal atopy	1.56	(1.14, 2.12)	0.0054
Older sibling	0.95	(0.7, 1.3)	0.75
Antibiotics by 1 year	2.25	(1.55, 3.27)	2E-05
Birthweight Z-score	1	(0.85, 1.17)	0.98

Nitrogen dioxide IQR	1.08	(0.66, 1.77)	0.76
------	--------------	------

New Supplementary Table 2. Clinical and environmental factors linked to the development of allergic diagnoses at 5 years.

- Figure 5C: upper and lower panel are not aligned.

Thank you for pointing this out. We have aligned the heatmap panels in what is now Figure 6. In addition, please note that the revised figure (shown below) also includes all changes made on the recommendation of other reviewers' comments regarding our metabolomics analysis and includes the new significant associations with butyrate and SCFA-producing microbes. Thank you again for all of your helpful feedback.

a Microbiota-age at 1 year: Youngest Oldest
PERMANOVA (2.2% v.e., p<0.001)

- Module 1** Amino acid and derivatives metabolism
- Module 2** Citric acid cycle
- Module 3** Steroid biosynthesis
- Module 4** Glycine and serine metabolism
- Module 5** Pyruvate metabolism
- Module 6** Sphingolipid metabolism
- Module 7** Lipid metabolism
- Module 8** Fatty acid biosynthesis
- Module 9** Spermidine and spermine biosynthesis
- Module 10** Lysophospholipid metabolism
- Module 11** Phosphotidylcholine metabolism
- Module 12** Short-chain fatty acid derivative metabolism
- Module 13** Oxidocarboxylic acid metabolism
- Module 14** Dicarboxylic acid metabolism

b

c

Reviewer #2 (Microbiome maturation and allergy).

Comments:

1. There is a long-standing debate whether all so called atopic diseases which have very distinct clinical features (skin or upper and/or lower respiratory tract involvement with and without concomitant allergic sensitization to food and/or aeroallergens) can be lumped together or whether they show divergent underlying pathomechanisms. One feature that may be common to many but not all of them is allergic sensitization which was assessed with a low cut-off (skin prick test ≥ 2 mm after subtraction of the negative control). Given that in the overall cohort “The majority of parents reported a history of ever having allergies and atopic diseases (77.0%)...”(Reference 45) such enrichment for a family history of allergies may have contributed to a high proportion of allergic sensitization in this population which may be the common feature of all assessed allergic outcomes. The authors should therefore show in the Venn diagram in supplemental figure 1 the additional overlap with allergic sensitization. If there is a strong overlap, then it will be very hard to disentangle the single clinical disease entities from allergic sensitization and the true association with delayed microbiota maturation may be with allergic sensitization rather than all the mentioned diseases. Given the smaller sample size (n=589) with the MetaCyc and metabolomics data, the number of subjects with clinical diagnoses without allergic sensitization will likely be too small to allow robust statistical analyses. Alternatively, stratification of analyses by allergic sensitization at any of the three assessed time points may help, but may suffer from the same limitation.

Thank you for this comment. We agree with the reviewer, and the potential shared underlying pathophysiological mechanism is in part what inspired the study. Our aim was not to disentangle microbiome signals unique to disease entities but instead to identify common microbiome signals across all these diseases, strengthening the argument that they may all have similar origins in abnormal immune education during infancy. We have clarified this aim within our introduction in the Introduction at **Lines 77-80**.

The suggestion to parse out participants with allergic diagnoses by their atopic sensitization (as defined by positive SPT tests) is an excellent one. As requested, we have now included allergic sensitization (+SPT) in Supplemental Figure 1.

Within the clinical cohort (Supplemental Figure 1a), 59.4% of participants diagnosed with asthma also had a +SPT at one of their 1-, 3-, or 5-year visits, 55.8% of participants diagnosed with atopic dermatitis also had a +SPT at one of their 1-, 3-, or 5-year visits, 66.8% of participants diagnosed with allergic rhinitis also had a +SPT at one of their 1-, 3-, or 5-year visits and 91.2% of participants diagnosed with food allergy also had a +SPT at one of their 1-, 3-, or 5-year visits.

The numbers within the metagenomic cohort are quite similar to the larger clinical one (Supplemental Figure 1b). 54.4% of participants diagnosed with asthma also had a +SPT at one of their 1-, 3-, or 5-year visits, 59.0% of participants diagnosed with atopic dermatitis also had a +SPT at one of their 1-, 3-, or 5-year visits, 70.0% of participants diagnosed with allergic rhinitis also had a +SPT at one of their 1-, 3-, or 5-year visits and 89.3% of participants diagnosed with food allergy also had a +SPT at one of their 1-, 3-, or 5-year visits.

To look more closely at the underlying commonality of allergic sensitization within our diagnoses, we parsed out children with a 5-year diagnosis by whether they had a recorded +SPT response at any of the

1y, 3y, or 5y visits (Supplemental Figure 1c). We then compared microbiota maturation at 1 year, measured by predicted age. What we found was that a lower predicted age at 1 year accompanied a later allergic diagnosis, regardless of skin prick test response. We also detected no significant difference (p -value = 0.6774) in predicted age between children with a 5-year diagnosis based on their SPT responses.

To look at overlap in another way, we also compared only children with a single 5-year diagnosis ($n=251$ children) to children with no diagnosis (Supplemental Figure 1d). We found that children with a single diagnosis had a significantly lower predicted age than children with no diagnosis (p -value = 0.00232).

We have now referenced these new analyses within the manuscript within **Lines 118-120**.

Additionally, please see below our new **Supplemental Figure 1**.

We thank the reviewer for their insightful comments, and we feel these additional analyses significantly improve our overall findings. We acknowledge that disentangling different signals of the allergic diseases based on underlying allergic sensitization is worthwhile and have noted this as a future direction in **Lines 372-377** in the Discussion section.

2. Any underlying other feature common to all allergic outcomes such as a family history, breastfeeding and antibiotic usage as shown in figure 1 A/B or persistent sensitization as defined in the LCA in reference 45 rather than any sensitization may be the true factor associated with microbiome maturation.

In this case, mediation analyses between the potential underlying tertium comparationis and the outcomes would be more appropriate.

Thank you for this interesting comment. We agree that both the underlying features during early life common to allergic outcomes play a clear role in microbiome development, and that proneness to persistent sensitization may also be reflective of microbiome maturation. Within this study, we focused on studying the relationship between maturation and allergic diseases at 5 years. We were not powered to limit our analyses to persistent sensitization at 1y, 3y, and 5y and still perform shotgun metagenomics and metabolomics.

To address your comment, we compared the effects of early-life exposures and microbiome maturation on the development of a 5-year allergic disease and have now included this in Supplementary Fig. 2. Similar to our previous Figure 1 (now current Figure 2) we performed a conditional logistic regression that included microbiota-predicted age in addition to known clinical/environmental features associated with allergic sensitization. Predicted age remained significantly protective against the development of one or more allergic diseases with adjustment for other covariates. In this multivariable model, maternal atopy and antibiotic usage remained significant risk factors. This was similarly the case for breastfeeding. The relationship between breastfeeding and allergies was reduced in the smaller, less powerful shotgun metagenomic cohort without predicted age included (aOR=0.74 (0.41, 1.32); p-value = 0.3) and this remained the same when predicted age was added.

While our current available clinical and environmental variables cannot explain the observed reduction in predicted age, we agree that further exploration of linking specific environmental influences and/or clinical diagnoses with microbiota maturation is important. We added a comment to this effect on **Lines 376-378** in the Discussion section.

3. Further along this line of reasoning: T which life style factors (family history, breastfeeding, antibiotic use) determined the 9 taxa associated with allergy development and protection, respectively?

Thank you for this inquiry.

Within the subset of metagenomics samples used within this study, the following are the results of models between microbes and antibiotic usage, breastfeeding, and family history (no species were significantly associated with maternal atopy).

Species	Variable	Coefficient	St. err.	P-Value	FDR
Megasphaera micronuciformis	Breastfeeding at 6 months	0.71	0.10	0.00	0.00
Veillonella atypica	Breastfeeding at 6 months	0.64	0.11	0.00	0.00
Tyzzereella nexilis	Breastfeeding at 6 months	-0.55	0.11	0.00	0.00
Clostridium innocuum	Breastfeeding at 6 months	-0.45	0.10	0.00	0.00
Intestinibacter bartlettii	Breastfeeding at 6 months	-0.37	0.10	0.00	0.00
Ruminococcus gnavus	Breastfeeding at 6 months	-0.33	0.09	0.00	0.00
Sellimonas intestinalis	Breastfeeding at 6 months	-0.41	0.12	0.00	0.00

Erysipelatoclostridium ramosum	Breastfeeding at 6 months	-0.29	0.11	0.01	0.01
Eggerthella lenta	Breastfeeding at 6 months	-0.20	0.07	0.01	0.02
Enterococcus faecalis	Breastfeeding at 6 months	-0.22	0.09	0.02	0.03
Escherichia coli	Breastfeeding at 6 months	0.29	0.12	0.02	0.03
Gordonibacter pamelaee	Breastfeeding at 6 months	-0.19	0.09	0.04	0.07
Ruminococcus bromii	Breastfeeding at 6 months	-0.25	0.12	0.04	0.07
Clostridium innocuum	Paternal atopy	-0.60	0.24	0.01	0.05
Erysipelatoclostridium ramosum	Paternal atopy	-0.56	0.25	0.02	0.07
Bifidobacterium longum	Antibiotic usage by 1 year	-1.54	0.40	0.00	0.00
Tyzzarella nexilis	Antibiotic usage by 1 year	1.01	0.27	0.00	0.00
Clostridium innocuum	Antibiotic usage by 1 year	0.73	0.25	0.00	0.01
Veillonella atypica	Antibiotic usage by 1 year	-0.80	0.27	0.00	0.01
Sellimonas intestinalis	Antibiotic usage by 1 year	0.81	0.29	0.01	0.02
Hungatella hathewayi	Antibiotic usage by 1 year	0.78	0.30	0.01	0.02
Megasphaera micronuciformis	Antibiotic usage by 1 year	-0.53	0.24	0.03	0.06

Supplementary Table. 4. Microbiota associated with important clinical features in allergic disease. MaAsLin2 results indicating the microbe identified as significant, the variable and comparison group, coefficient of association, standard deviation, p-value, and FDR-corrected p-value.

While none of these features accounted for all the bacterial alterations that we observed in relation to both allergic disease diagnoses and predicted age, we have added these results in Supplementary Fig. 4 as well as Lines 188-192 in the Results section of the manuscript.

We believe that this analysis provides meaningful context to our original findings, as it relates the microbiome species' signals back to participant clinical and familial information. We thank the reviewer for this question and appreciate that improvement to our manuscript as a result of these additional analyses.

4. The CHILD cohort followed 3495 enrolled subjects. The authors must report if any selection bias, e.g. further enrichment of a positive family history or of allergic sensitization/outcomes occurred in the subsample of n=589.

Thank you for this important comment. The general cohort of the CHILD study included 3,264 infants eligible at birth. In order to achieve our aim of creating a clear comparator group we did exclude individuals if they did not have a 5-year diagnosis *AND* a recorded negative SPT response *AND* no history of wheeze data for all three of the visits (1y, 3y, and 5y). For example, this meant that if a child was considered non-allergic by physician diagnosis at 5 years but had a positive SPT response at 1 year, then they were excluded. Furthermore, we required documented SPT and wheeze data for every visit (1y, 3y, and 5y), so this meant that children with no diagnosis and no history of allergic sensitization or wheeze

could still be eliminated if they had missed a single visit. Because of this stringent selection, the clinical cohort used in our study is understandably different from the larger general cohort as we eliminated a number of sub-clinical atopic children. Furthermore, the requirement to attend every visit also likely introduced some bias towards families with the available time and resources to achieve perfect adherence to the study.

Based on your suggestion, we have now added Supplemental Table 2. When comparing the clinical cohort in our study to the larger general cohort, we measure a difference in breastfeeding (4% increase) and maternal atopy prevalence (1.2% increase). Importantly, the samples in metagenomic and metabolomic analysis did not significantly differ from our clinical analysis. We have now added this as Supplementary Table 1 (Line 140).

Variable	Overall CHLD Population	Clinical	p value (Clinical vs. Overall)	Metagenomic	p value (Metagenomic vs. Clinical)	Metabolomics	p value (Metabolomics vs. Metagenomic)
No. patients	3264	1115		589		509	
Male, n(%)			0.68		0.57		0.9
	1717 (52.6%)	595 (53.4%)		323 (54.8%)		281 (55.2%)	
Ethnicity of Child, n(%)			0.35		1		0.85
Caucasian White	2043 (63.6%)	689 (62.1%)		365 (62.1%)		312 (61.4%)	
Non-Caucasian	1167 (36.4%)	421 (37.9%)		223 (37.9%)		196 (38.6%)	
Delivery Mode, n(%)			0.79		0.57		0.99
Vaginal	2412 (74.8%)	814 (74%)		421 (71.7%)		363 (71.6%)	
C-Section with labor	425 (13.2%)	146 (13.3%)		87 (14.8%)		74 (14.6%)	
C-Section without labor	387 (12%)	140 (12.7%)		79 (13.5%)		70 (13.8%)	
Breastfeeding status at 6 months, n(%)			0.0091		0.95		1
	2323 (76.4%)	884 (80.2%)		473 (80.4%)		408 (80.3%)	
Season of birth, n(%)			0.85		0.79		0.9
Spring	889 (27.2%)	293 (26.3%)		165 (28%)		146 (28.7%)	
Summer	830 (25.4%)	297 (26.6%)		145 (24.6%)		130 (25.5%)	
Fall	755 (23.1%)	259 (23.2%)		137 (23.3%)		120 (23.6%)	
Winter	790 (24.2%)	266 (23.9%)		142 (24.1%)		113 (22.2%)	
Atopy of father, n(%)			0.84		1		0.94
	1663 (67.7%)	622 (67.2%)		330 (67.3%)		289 (67.7%)	
Atopy of mother, n(%)			0.068		0.29		1
	1727 (57.7%)	668 (60.9%)		371 (63.5%)		321 (63.6%)	
Having older sibling, n(%)			0.89		0.3		0.86
	1452 (45.9%)	500 (45.6%)		282 (48.3%)		247 (49%)	
Antibiotics use in the first year of life, n(%)			0.11		0.8		0.65
	605 (18.5%)	231 (20.7%)		125 (21.2%)		102 (20%)	
NO2 in the first year of life			0.23		0.33		0.87
Median (Range)	9.1 (0.5, 8.8, 30.5)	(1.2, 29.1)		9.1 (1.2, 29.1)		9 (1.2, 29.1)	
IQR (Q1,Q3)	4.6, 13.3	4.5, 12.9		4.7, 13.3		4.6, 13.2	
Birth weight Z-score			0.76		0.47		0.74
Median (Range)	-0.1 (-3.1, 4.3)	-0.1 (-3.1, 3.7)		-0.1 (-2.6, 3.7)		-0.1 (-2.6, 3.7)	
IQR (Q1,Q3)	-0.7, 0.5	-0.7, 0.6		-0.7, 0.6		-0.7, 0.7	

Again, we thank the reviewer for flagging the importance of this comparison. We have also added a note in the Discussion section at Lines 345-349 outlining the inherent differences between the CHILD cohort and our analysis sub-cohorts.

5. The methods section does not report allergic sensitization to pollen, yet in reference 45 SPTs to also aeroallergens are mentioned for age 3 years. How many children were ONLY sensitized to tree and/or grass but not to other allergens ? In other words, how many children would have been non-atopic at age 3 years when excluding the children ONLY sensitized to tree and/or grass ?

Thank you for pointing this out. We apologize for the confusing language. We have updated the methods to replace the original term “environmental inhalant allergens” to include each pollen source in the Methods section at Lines 416-420 to be more specific.

6. The authors report a decrease in Shannon diversity at age 3 and age 12 month – was such association also seen for other measures of alpha diversity ?

Thank you for this inquiry. The following analyses are our reported changes of Observed and Faith’s PD alpha diversity.

3-month sample diversity stratified by allergic disease diagnoses at 5 years:

Diversity measure	Allergic disease	Mean	P-value
Observed			
	Healthy controls	17.81	Ref
	Atopic dermatitis	17.10	0.93
	Asthma	16.43	0.39
	Allergic rhinitis	17.55	0.64
	Food allergy	16.58	0.48
	One or more	17.22	0.56
	Two or more	16.65	0.03
Faith’s PD			
	Healthy controls	4.02	Ref
	Atopic dermatitis	3.94	0.94
	Asthma	3.93	0.94
	Allergic rhinitis	4.02	0.57
	Food allergy	3.85	0.47
	One or more	3.97	0.80
	Two or more	3.92	0.16

Notably, alpha diversity is consistently not significant across all comparisons, except from two or more allergic diseases for observed diversity, for the samples collected at 3 months, when looking at these two additional measures of alpha diversity.

1-year sample diversity stratified by allergic disease diagnoses at 5 years:

Diversity measure	Allergic disease	Mean	P-value
Observed			
	Healthy controls	37.7	Ref
	Atopic dermatitis	34.8	0.0013
	Asthma	34.6	0.005
	Allergic rhinitis	35.5	0.026
	Food allergy	34.35	0.013
	One or more	35.27	0.0015
	Two or more	34.27	0.0066
Faith's PD			
	Healthy controls	7.05	Ref
	Atopic dermatitis	6.71	0.0037
	Asthma	6.70	0.011
	Allergic rhinitis	6.79	0.046
	Food allergy	6.67	0.028
	One or more	6.77	0.0048
	Two or more	6.63	0.0067

In contrast to the results of analyses of the microbiota communities at 3 months, the significance reported in the original analyses using Shannon diversity continues for a number of comparisons using Observed and Faith's alpha diversity. This includes significance between the healthy subset of participants and participants with atopic dermatitis, asthma, one or more diagnoses, and two or more diagnoses at 5 years. In addition, the 1-year samples of participants with two or more diagnoses at 5 years are also significant for differences in Faith's PD alpha diversity. This is interesting in that having two or more allergic diagnoses at 5 years corresponds to the most severe alterations in diversity at 1 year.

Overall, these additional comparisons support our further exploration of alterations within the 1-year samples as they reflect the biological imbalances that may be a source or a reflection of immune system dysregulation.

7. Most of the manuscript focuses on the maturation at age 12 months. The investigators however also collected fecal samples at age 3 month. Was there any association between microbiome features, in particular alpha-diversity at age 3 month at the maturation as assessed by age 12 month ?

We thank the reviewer for this thoughtful question. We performed this analysis with the 3-month samples included in this study. Although it is not significant, 3-month Shannon diversity tends to be associated with predicted age by age 12 months, with adjustment for exact age and random factor of collection site ($p = 0.0553$).

	Value	St. Error	DF	t-value	p-value
(Intercept)	0.9086076	0.02350310	604	38.65906	0.0000
Shannon	0.0204377	0.01064297	604	1.92030	0.0553
exact_age	0.0164659	0.06521477	604	0.25249	0.8008

While this link between 3-month sample diversity and 1-year sample predicted age is interesting, we have opted not to include these statistics within the manuscript to instead focus on results with significance. We again thank the reviewer for this thoughtful question and hope that these findings are sufficient answers to their inquiry.

8. According to figure 5c tryptamine measured in fecal samples was strongly positively correlated to most of the 11 Metacyc-annotated gene pathways. Yet, in figure 6 they appear as independent contributors in the structural equation modeling diagram. How can individual independent effects be disentangled for each individual contributing factor ? This also applies to the discussion section where much emphasis is put on the trace amines.

Thank you for this comment. The fact that the trace amines were strongly correlated with the Metacyc-annotated pathways indicated that they were acting in a similar direction within the gut. We interpreted this as them being involved in an increased risk for allergic disease due to decreased predicted age and hypothesized that they were working in concert to effect or reflect an imbalance in the gut. As SEMs are based upon hypotheses, we hypothesized that, together, these features were mediating the relationship between microbiota-predicted age at 1 year and the development of allergic disease at 5 years.

Thus, within the structural equation model, we combined the pathways and metabolites into a latent variable using confirmatory factor analysis, which allows the researcher to test the hypothesis that a relationship between observed variables and their underlying latent constructs. At its core, the purpose of latent variables is to translate that multiple observed variables are potentially imperfect manifestations of one underlying cause. Given the fact that microbiome data is interrelated and sparse, the combination of features may therefore offer a more appropriate underlying representative variable across the proposed variables.

To further clarify this, we provide a sensitivity analysis with an SEM including only the metabolites and not the Metacyc-annotated pathways. Moreover, we have now included butyrate within our SEM due to its association with microbiome features in the revised correlation heatmap, as well as its biological relevance to the bacteria that were depleted in participants who went on to be diagnosed with allergic diseases at 5 years.

When only using the trace amines tryptamine, tyramine, and phenylethylamine, and butyrate, the following is the output of the SEM lavaan model:

Comparative Fit Index (CFI) 0.855

Defined Parameters:

	Estimate	Std.Err	z-value	P(> z)	Std.lv	Std.all
indirect_all	-0.200	0.113	-1.773	0.076	-0.200	-0.023
direct_all	-1.775	0.543	-3.271	0.001	-1.775	-0.205
total_all	-1.976	0.531	-3.723	0.000	-1.976	-0.229

The latent variable including only the metabolomic features tends to be a significant mediator. Again, given that the indirect effect of metabolites is not powerful enough to be significant by themselves, we continue to deem the integration of the metabolites with the significant pathways as a meaningful mediating factor within the current manuscript.

We found that in addition to these Metacyc-annotated pathways, the indirect effect is significant with a high comparative fit index.

Comparative Fit Index (CFI) 0.949

Defined Parameters:

	Estimate	Std.Err	z-value	P(> z)	Std.lv	Std.all
indirect_all	-1.825	0.523	-3.491	0.000	-1.825	-0.211
direct_all	-0.151	0.744	-0.203	0.839	-0.151	-0.017
total_all	-1.976	0.531	-3.723	0.000	-1.976	-0.229

Thus, we have continued to include the metabolites of interest as identified within our analyses, as well as the functional pathways that may be underlying the relationship between the species-derived predicted age of the samples.

9. Line 185: I assume this relates to 12 mo fecal samples ?

Yes, our follow-up analyses are performed using the 12-month fecal samples. We have clarified this on Lines 172 - 175 prior to these points.

Reviewer #3 (Remarks to the Author):

Hoskinson et al have studied the differences in gut microbiota between children with allergic disease and compared them to a control group. The study is interesting and creative in the way it has addressed the problem and I think will be of interest to Nature Comms readers. I very much enjoyed reading it.

We thank the reviewer for their positive feedback.

I have concentrated on the metabolomics analysis as my area of expertise. The reasoning of the paper is clear, although more details on how the technical aspects of metabolomics may be influencing the results should be considered.

The conclusions have been drawn on a single 2 mL aliquot of stool. This is common practice in this field, but there is good evidence to suggest that certain metabolites may not be homogeneously spread throughout the stool. In addition, the methods for stool preparation are not well described, but according to their cited paper Moraes et al, may have been refrigerated for up to three days. This will certainly affect the metabolome, and may also have affected the microbiome. Was any attempt made to assess and control for length of refrigerated storage?

Regarding stool homogeneity: Stool was scooped from diapers into a single collection tube that was then aliquoted into smaller freezer tubes depending on the amount prior to long-term storage by research staff without any homogenizing steps.

Regarding short-term storage: We thank the reviewer for this insightful comment. We had not accounted for storage preprocessing steps prior to the reviewer's comment, but we completely agree that it could introduce technical variability into our results. We appreciate that our new analysis accounting for the time between collection and long-term storage substantially improves the validity of our findings. Between sample collection and long-term -70°C storage, samples were placed at 4°C for varying lengths of time, and this period of short-term storage was recorded by CHILD study investigators and staff.

Our initial analysis incorporating this storage processing time can be seen in the following figure. To summarize, when we correlated Predicted age and processing time, we saw no association (panel a: spearman rho= -0.13, p=0.76). While microbiome DNA abundances would be much slower to shift at 4°C, we wanted to alleviate your (and subsequently our) concerns that processing time had not resulted in artifacts when quantifying either the microbiome-derived predicted age or differences in 5-year allergy diagnosis.

We also interrogated whether there were any differences in processing time between infants who would receive a 5-year allergy diagnosis or not (panel b and c). Encouragingly, we found no significant difference between the processing times for our primary groups of interest, the healthy, non-allergic controls and participants with one or more allergic diseases (p = 0.44). Further, as seen in the histogram, the vast majority of samples were placed into long-term in the first 24-36 hours, but our data shows a handful of samples that were kept in short-term storage for up to 72 hours and one in particular that extended to 96 hours.

As you rightfully point out, metabolite concentrations could change due to volatility or breakdown in this period, so we looked at the effect of processing time on metabolites using our Maaslin2 analysis (while continuing to adjust for exact age and treating the site as a random effect). Similar to what has been shown in other literature, processing time does indeed affect concentrations of certain metabolites as shown below.

Because of this, we have now added processing time to the variables that we adjust for in all of our analyses that include metabolic profiles (current Figure 6 and Figure 7). This has also been added to our manuscript in Lines 241-243.

We have also clearly outlined our steps within our revised Methods section at Lines 490-507.

Encouragingly, when we adjust for processing time, our results stayed consistent. Our primary findings remain. Furthermore, thanks to the reviewer's suggestion, we now observe a significant association between Butyrate (a volatile SCFA that is sensitive to processing time but has been linked to a number

of important biological processes, including allergies) and our microbiome features associated with predicted age. Our newly generated figures can be viewed within our revised manuscript Fig. 6 and in Supplemental Fig. 6, both shown below.

Fig. 6. Relating significant microbiome features with metabolic profiles in the gut. **a** Principal component analysis (PCA) plot of variance within the 1-year gut metabolome and colored by predicted age distribution. **b** Weighted gene co-expression analysis (WGCNA)-determined modules and

interactions of metabolites in the 1-year gut, mapped using Cytoscape. **c** Spearman correlation heatmap of the relationship between metabolites, WGCNA clusters, and microbiome features of interest identified in Fig. 3 and 4. (*) $q < 0.05$.

Supplementary Fig. 6. Metabolites linked to predicted age and allergic disease. MaAsLin2 results of metabolites associated with **a** predicted age and **b** one or more allergic disease.

In addition, we have adjusted the text of the manuscript to reflect our findings, ranging from **Lines 231 – 284**.

What data have you to show that your method gives reliable, repeatable results?

In addition to our robust CHILD cohort dataset, including $n = 204$ healthy control samples and $n = 305$ case samples, including participants with atopic dermatitis ($n = 182$), asthma ($n = 90$), food allergy ($n = 65$), and allergic rhinitis ($n = 96$), all metabolic processing was performed by The Metabolomics Innovation Centre (TMIC) which is a nationally funded metabolomics core facility spanning 4 Canadian Universities (<https://metabolomicscentre.ca>) that has contributed to hundreds of peer-reviewed publications. The methods used in our analyses have thus been previously published¹⁻⁴. Both NMR and LC-MS assays are validated for fecal samples. The core also cross-checks the concentration values for overlapping metabolites by these two orthogonal methods, confirming the accuracies of the reported

concentration values. As a measurement of reliability, TMIC verified assay precision by calculating the coefficient of variability (CV%) for both NMR (CV< 5%) and LC-MS/MS (CV<10%) analyses.

This has been updated within the methods section in **Lines 466-474**.

The metabolomics protocol does not contain enough information to allow the experiment to be repeated. Specifically, it does not give any information on extraction methods. There should be more details provided in the supplementary on the extraction techniques and the metabolites which were detected by NMR and MS and whether any overlapped. The results cannot be reliably assessed without more information on number of batches, inter and intra batch effects and assessment of technical biases in the datasets.

Thank you for this comment and for the detailed consideration of the metabolomics methods. We have worked with TMIC to amend our main methods section with additional details of the primary methodology for our metabolomics analysis. We have also added comprehensive details to the supplementary information of the metabolomics data collection.

We have outlined our responses in our point-by-point responses to the reviewer's comments below and have noted the lines in which we have altered the main Methods and Supplementary Files of metabolomics protocols.

Supp methods section:

For Stool preparation methods, please include the following details:

Number of batches run for each method

62 batches for NMR; 27 batches for LC-MS/MS (**Lines 466-474**).

Batch differences – how they were assessed and corrected

We had originally adjusted for batches within our Maaslin2 analysis, but have since improved our batch correction methods based on your comments. We now use the R package ComBat to separately correct the NMR dataset and the LC-MS/MS dataset by their respective batches. This is outlined within the main text and **Lines 490-507**.

Carryover or background levels of metabolites – how were they assessed and corrected for.

Whether or not there was a significant carry-over was observed by comparing the Cal7 injection and the following double blank injection for LC-MS/MS data evaluation while background subtraction methods were not used for NMR, as blanks were checked properly before analyzing the samples (**both points added to the Supplementary Methods file**).

Number of metabolites targeted with each method

Number found for each method

Number which were measured in the quantitative range for each method.

31 metabolites were targeted for NMR; and 590 metabolites were targeted for LC-MS/MS, and we received data for each of these metabolites from TMIC including whether values fell below the limit of detection (LOD).

All of the NMR metabolites were used for downstream statistical analyses.

For LC-MS/MS: We eliminated 244 metabolites that were detected in less than 20% of either 3 month or 1-year samples. Additionally, we eliminated another 132 metabolites with low variance defined by standard deviation of less than 0.005.

This resulted in 245 total metabolites used for our downstream analyses (31 detected via NMR and 214 detected via LC-MS/MS). This is outlined within the main text and Lines 490-507.

LC-MS analysis supp method

Red PEEK tubing – this is not very informative since different companies use different colours to denote different internal diameters. Please include company and internal diameter and length of tubing.

Thank you for your inquiry, the tubing information can be found below (and in the LC-MS/MS Supplementary Methods file):

Company: Sigma-Aldrich
O.D. × I.D. 1/16 in. × 0.005 in
Length: 50 cm

LC-MS analysis – please give more details about the calibration concentration ranges, number of calibration points and whether standards were run as individual standards or as mixed standard calibrants. What was the matrix that the calibrants were analysed in and how was background and carryover checked and accounted for.

We thank the reviewer for their comment. The following are more details of the calibration concentration ranges, the number of calibration points, and the subsequent analysis of matrix calibrants and background carryover (added to Supplementary Methods file).

Firstly, there were 7 calibration points and the calibrants were run as a mixture. Calibrants were prepared in a matrix-free solution. Whether or not there was a significant carry-over was observed by comparing the Cal7 injection and the following double blank injection. The calibration ranges were as follows:

Metabolite	Calibration range
1,3-Diaminopropane	0.15~12
1-Methylnicotinamide	1.875~150
3-Methoxytyramine	0.2~16
5-Hydroxylysine	0.75~60
5-Methoxytryptamine	0.05~4
5-Methyluridine	0.25~20
7-Methylguanine	0.5~40
AABA	0.8~64
Adenine	0.125~10
Adenosine	0.5~40
Agmatine	0.1~8
Ala	200~16000
Allantoin	3.75~300
alpha-Aminoadipic acid	2.5~200
Arg	15~1200

Asn	25~2000
Asp	30~2400
beta-Alanine	2.5~200
Betaine	5~400
Carnosine	0.2~16
Choline	2~160
cis-4-Hydroxyproline	0.6~48
Citrulline	30~2400
Creatine	1~80
Creatinine	6~480
Cystathionine	1~80
Cytidine	1.6~128
Cytosine	1~80
Deoxyadenosine	1~80
Deoxycytidine	1.5~120
Deoxyguanosine	1.6~128
Deoxyinosine	1.5~120
Deoxyuridine	4~320
Diacetylspermine	0.3~24
Dimethylamine	3~240
DOPA	0.3~24
Dopamine	0.2~16
Epinephrine	0.1~8
Ethanolamine	12.5~1000
GABA	3.2~256
Gln	100~8000
Glu	250~20000
Gly	125~10000
Guanine	0.5~40
Guanosine	2~160
His	12~960
Histamine	0.2~16
Homoarginine	0.6~48
Homocitrulline	2.5~200
Hypoxanthine	28~2240
Ile	12.5~1000
Indole	0.25~20
Indole-3-acetamide	0.2~16
Inosine	1.6~128
Kynurenine	0.5~40
Leu	20~1600
Lys	200~16000
Met	40~3200
Methionine sulfoxide	5~400
Methylamine	7.5~600
Methylhistidine	4~320
N1-Acetylspermidine	1~80
N2-Acetyl-Orn	0.5~40

N-Acetylputrescine	7.5~600
Nicotinamide ribotide	3~240
Nitro-Tyr	0.625~50
Norepinephrine	0.15~12
Nudifloramide	1.5~120
Orn	10~800
Phe	50~4000
Phenylethylamine	0.1~8
Pro	30~2400
Putrescine	0.5~40
Sarcosine	0.2~16
Ser	50~4000
Serotonin	0.1~8
Spermidine	1.35~108
Spermine	0.25~20
Taurine	2~160
Thr	50~4000
Thymidine	5~400
Thymine	0.4~32
TMAO	1~80
total DMA	2~160
trans-4-Hydroxyproline	0.6~48
Trimethylamine	5~400
Trp	12.5~1000
Tryptamine	0.5~40
Tyr	25~2000
Tyramine	0.5~40
Uracil	25~2000
Uridine	4~320
Val	100~8000

Xanthine	25~2000
Uric acid	20~1600
p-Cresol sulfate	3~240
4-Ethylphenyl sulfate	0.125~10
Indoxyl sulfate	4.5~360
Lactic acid	12.5~1000
3-Aminoisobutyric acid	4~320
Dimethylglycine	0.625~50
2-Hydroxybutyric acid	0.5~40
2-Hydroxyisobutyric acid	0.5~40
3-Hydroxybutyric acid	0.25~20
3-Hydroxyisobutyric acid	0.2~16
Glyceric acid	2~160
Guanidoacetic acid	12.5~1000
N-Acetyl-Gly	4~320
2-Hydroxy-2-methylbutyric acid	1.25~100

2-Hydroxyisovaleric acid	0.3125~25
3-Hydroxyisovaleric acid	0.5~40
3,4-Dihydroxybutyric acid	6~480
Benzoic acid	0.5~40
5-Oxoproline	250~20000
Pipecolic acid	1.6~128
Guanidinopropionic acid	0.3~24
N-Acetyl-Ala	10~800
2-Hydroxy-3-methylvaleric acid	0.5~40
Phenylacetic acid	25~2000
Threonic acid	3.75~300
4-Hydroxybenzoic acid	0.625~50
Salicylic acid	0.8~64
N-Acetyl-Ser	1.25~100
Xanthosine	0.5~40
2-Hydroxyphenylacetic acid	2.5~200
3-Hydroxyphenylacetic acid	5~400
4-Hydroxyphenylacetic acid	3~240
Orotic acid	1.5~120
N-Acetyl-Pro	2~160
Tiglylglycine	0.8~64
N-Acetyl-Val	0.5~40
Indoxyl glucoside	2.5~200
Indole-3-carboxylic acid	0.5~40
Quinoline-4-carboxylic acid	0.25~20
N-Acetyl-Leu	0.5~40
Quinaldic acid	0.25~20
N-Acetyl-Ile	6.25~500
Shikimic acid	0.8~64
N-Acetyl-Asn	2~160
3-Indoleacetic acid	25~2000
Argininic acid	2.5~200
Hippuric acid	5~400
Caffeic acid	1.25~100
Homovanillic acid	2.5~200
HPPHA	0.25~20
N1-Acetyl-Lys	2.5~200
N6-Acetyl-Lys	6.25~500
N-Acetyl-Gln	5~400
Kynurenic acid	0.5~40
Indole-3-propionic acid	0.5~40
5-HIAA	1.6~128
N-Acetyl-Met	0.5~40
2-Methylhippuric acid	0.25~20
cAMP	0.25~20
p-Hydroxyhippuric acid	1.25~100
N-Acetyl-His	1.25~100
Indolelactic acid	0.5~40

N-Acetyl-Arg	5~400
Pyruvic acid	2.5~200
N-Acetyl-Tyr	1.25~100
Oxalic acid	10~800
Acetoacetic acid	12.5~1000
Malonic acid	1.5~120
N-Acetyl-Trp	0.625~50
alpha-Ketoisovaleric acid	0.6~48
Fumaric acid	5~400
Maleic acid	5~400
Methylmalonic acid	1~80
Succinic acid	12.5~1000
Phenylacetylglutamine	5~400
2-oxoisocaproic acid	2.4~192
5-Aminolevulinic Acid	2.5~200
Ethylmalonic acid	0.5~40
Glutaric acid	6.25~500
Malic acid	12.5~1000
N-Methyl-Asp	5~400
2-hydroxyglutaric acid	3.125~250
Tartaric acid	5~400
FDCA	0.5~40
2-oxoadipic acid	12.5~1000
3-Methyladipic acid	1.25~100
3-Deoxyglucosone	6.25~500
Quinolinic acid	0.5~40
Indoxyl glucuronide	0.5~40
N-Acetyl-Asp	5~400
4-Hydroxyphenylpyruvic acid	1~80
N-Acetyl-Glu	10~800
CMPF	0.125~10
Oxalacetic acid	50~4000
alpha-Ketoglutaric acid	2.5~200
cis-Aconitic acid	1.25~100
Isocitric acid	2.5~200
Citric acid	10~800

How was identification carried out?

Individual standards were used for all reported metabolites. Identifications were done with respect to standards and MRM transitions (added to Supplementary Methods file).

DFI – unexplained acronym

DFI stands for direct flow injection (added to Supplementary Methods file).

NMR analysis supp method

“Typically all of visible peaks were assigned” – this is described in the methods as a targeted method, but this appears to be untargeted. Please clarify.

We apologize for the awkward wording. The NMR analysis indeed satisfies the term “targeted”; the spectra were profiled against a known metabolite library. Regarding the assignment of all visible peaks, this applies to when, within this targeted assay, very low abundant peaks are not visible for manual assignment and are therefore not reported (as those are hard to quantify via a targeted method) (added to Supplementary Methods file) and updated within the Methods section **Lines: 476-488**.

Also give an indication of background subtraction methods if used, how many peaks were discovered in the average sample and what level of matching similarity was required for identification.

Background subtraction methods were not used, as blanks were checked properly before analyzing the samples (added to Supplementary Methods file).

Main paper:

1) How was this selection of stool samples chosen: randomly, a select group or was there any known bias in the selection.

Thank you for this inquiry. For this paper, any participants from the clinical analyses in Figures 1 and 2 that had available metagenomic or metabolomic data were analyzed. We did not apply any additional selection. This is reflected in **Supplemental Table 1** demonstrated that they had similar demographics to the clinical analysis used in this paper.

The full metagenomic cohort was recently published in *Med* and contains 3 month and 1 year sequencing data from nearly 1500 infants ⁵. All of those sequenced stool samples were also sent to TMIC for metabolic profiling, with the only eliminating criteria being not enough sample mass or the presence of diaper fibers. Any subsequent metagenomic or metabolomic data from participants used in the clinical analysis from Figures 1 and 2 were analyzed for Figures 3-7.

2) How much stool was used, and what efforts were made to make sure it was a representative sample, especially in the original collection period.

During the original collection period, stool was scraped off of diapers and no additional homogenization of the stool sample was performed. We have now added an additional sentence within the discussion on **Lines 354-360** to clarify this for readers.

For the metabolomics analysis, steps to make sure it was a representative sample included powdering the feces in liquid nitrogen and quickly transferring the samples to individual Eppendorf tubes, this was noted in the NMR and LC-MS/MS protocols. Six hundred microliters of ice-cold HPLC water were added to the fecal powder (60-65 mg) and vortexed vigorously for 5 min. Then it was shaken for 25 min at 1000 rpm at 40C on a shaker followed by sonication at 4°C for 15 min. The sonicated sample tubes were centrifuged at 14000 rpm for 20 min at 40C or in the cold room. 500 uL of the supernatant was then transferred to the pre-washed 3 KDa cut-off centrifugal filter units (Amicon Microcon YM-3) and centrifuged at 11000 rpm for 20 min at 40C or in the cold room. A 200 uL filtrate was then transferred to a new Eppendorf tube and 50 uL buffer (54% D2O:46% 1.75 mM KH2PO4 pH 7.0 v/v containing

5.84 mM DSS (2,2-dimethyl-2-silcepentane-5-sulphonate), 5.84 mM 2-chloropyrimidine-5 carboxylate) was added to it.

3) The variance is relatively small, percentage wise. Given the inherent difficulties with achieving representative faecal samples, what measures were taken to validate this result with an independent cohort.

At this point in our work, we have not validated metabolite concentrations and associations within an independent cohort apart from CHILD; and we support this as a future investigative step. We have added additional lines within our Discussion section to highlight the necessity of such a study at Lines 356-362.

One note regarding the variance could be the fact that infants have a rather homogenous diet compared to older humans and thus may have less variation in their gut metabolomes.

Line 247: you make a correlation between number of significant metabolites found and relative importance of the pathways they map to. However, whether metabolite concentrations change can also be a function of rate limiting steps in the metabolic pathway. This is more difficult to measure in a mixed microbial environment, but was it considered when analysing the results?

Thank you for this interesting comment. This is a meaningful point in that metabolite concentrations can indeed change according to the slowest step in the related metabolic pathway, thereby determining the overall rate of the other reactions in the pathway and the metabolic output. We agree that it is worthwhile to parse out individual enzymatic steps with metabolite biosynthesis and breakdown. However, this is outside the scope of our study and also difficult to appropriately analyze using our current dataset. Rate-limiting steps control the rate of a series of biochemical reactions; however, it is likely that there are multiple steps that control the rate of each of the pathways measured and each controlling step controls the rate to varying degrees. This, in addition to the mixed microbial environment, makes it difficult to attribute individual rate-limiting steps using ‘omics’ technologies within the uncontrolled environment of the infant's gut. We did not intend to attribute changes in particular metabolites to pathways or metabolites, and we have to use careful language that reflects this. Instead, our aim was to identify specific metabolites that, together with alterations in the microbiome, might be indicative of an imbalanced gut environment and therefore influence the development of allergic disease. Using the associations identified within this study, we hope that future validation studies can link specific gut metabolites with species-specific enzymatic steps, but this is outside of the scope of this current study. To incorporate this point into the manuscript, we have now included a section on this topic within the discussion, beginning at Lines 368-380.

Line 411 unclear – “water content stool weight” – do you mean wet weight of stools?

Water content was measured by weighing the fecal samples before and after lyophilization, the mass difference of the two weights was considered as the amount of water. The unclear language has been updated to reflect that the concentrations were normalized to dry weight measured after lyophilization. This has been included in the paper in Lines 476-488.

Has the method been specifically validated for stool, and are the concentrations accurate, especially if the standards are run in a matrix free solvent.

Both NMR and LC-MS assays are validated for fecal samples. The core also cross-checks the concentration values for overlapping metabolites by these two orthogonal methods, confirming the accuracies of the reported concentration values. These details have been included in the manuscript's main Methods section at Lines 466-474, as well as in the Supplementary files

413: missing values can bias a dataset: how many missing values were there as a percentage of the dataset, and were they missing at random or missing not at random.

The original library of 590 targeted metabolites included a large proportion of lipid metabolites. Some of these metabolites (e.g. triglycerides) are naturally absent or found at trace levels in stool and so were detected below the limit of detection. Please see below piecharts. As far as other missing metabolites (we are defining missing as metabolites detected below the limit of detection), this was largely random and mostly relates to the interindividual variation between participants. To account for natural variability while still making sure that we were only analyzing metabolites whose effects could meaningfully be applied to the larger cohort, we required metabolites to be present in only 20% of either 3-month or 1-year samples or have variation of at least a standard deviation of $0.005\mu\text{mol/gram}$.

I think you have used a standard approach to absolute quantification, using an internal standard to correct for technical variation and then assessing the result against a calibration curve. However, the way it is written here sounds as if you are calculating a ratio and then assessing it against a non-ratio measurement to determine the concentration. Perhaps reconsider the wording.

Thank you for this comment. We have adjusted the wording to the following on Lines 476-488:

I think to publish in a journal such as Nature Communications, there is a clear expectation that data should be freely available on a site such as Metabolytes and not require the reader to contact the author. This should preferably include the raw data since this is important to assess batch effects, contaminations, incorrect identifications etc.

We have uploaded all of our NMR and LC-MS/MS data to MetaboLights and this will be made publicly available. The new accession number is MTBLS7919. Line 604.

Author contributions: it should be more clearly set out who carried out the metabolomics analyses.

We thank the reviewer for pointing this out. We have now adjusted Lines 793-795 to state who carried out these analyses.

“Investigation & Formal Epidemiological, Metagenomic, and Metabolomic Analysis, C.H, D.L.Y.D, and C.P.”

- 1 Vergara, A. *et al.* Urinary angiotensin-converting enzyme 2 and metabolomics in COVID-19-mediated kidney injury. *Clinical Kidney Journal* **16**, 272-284, doi:10.1093/ckj/sfac215 (2023).
- 2 Bridgman, S. L. *et al.* Childhood body mass index and associations with infant gut metabolites and secretory IgA: findings from a prospective cohort study. *International Journal of Obesity* **46**, 1712-1719, doi:10.1038/s41366-022-01183-3 (2022).
- 3 Drall, K. M. *et al.* Clostridioides difficile Colonization Is Differentially Associated With Gut Microbiome Profiles by Infant Feeding Modality at 3-4 Months of Age. *Front Immunol* **10**, 2866, doi:10.3389/fimmu.2019.02866 (2019).
- 4 Zheng, J., Zhang, L., Johnson, M., Mandal, R. & Wishart, D. S. Comprehensive Targeted Metabolomic Assay for Urine Analysis. *Analytical Chemistry* **92**, 10627-10634, doi:10.1021/acs.analchem.0c01682 (2020).
- 5 Azad, M. B. *et al.* Breastfeeding, maternal asthma and wheezing in the first year of life: a longitudinal birth cohort study. *European Respiratory Journal* **49**, 1602019, doi:10.1183/13993003.02019-2016 (2017).

REVIEWERS' COMMENTS

Reviewer #1 (Remarks to the Author):

The authors addressed all comments.

Figure 1A can remain as it is, but please keep also the table with the associated p-values and OR available as supplementary table to allow the reader to easily re-order the rows based on values. I recommend this manuscript for publication.

Reviewer #3 (Remarks to the Author):

The authors have made substantial efforts to revise their manuscript and I am pleased for them that it has resulted in an additional finding.

I have only one remaining issue: that of treatment of missing values. "Missing not at random" can refer to the distribution across samples, but it commonly is also used to describe the phenomenon that features with much lower average intensity are more likely to be missing than those with strong intensities. This can bias statistical results, especially where the missingness is more due to technical issues rather than true biological differences. The authors have not referred to this scenario in their analysis and therefore I don't know if they have considered it.

The authors may therefore like to do a sanity check and just ensure that none of their reported important metabolites are affected by this issue. I don't think that this should necessitate an additional round of review unless requested by the editor.

POINT-BY-POINT RESPONSE TO REVIEWER COMMENTS

Comments from Reviewer #1

Reviewer #1 (Remarks to the Author):

The authors addressed all comments.

Figure 1A can remain as it is, but please keep also the table with the associated p-values and OR available as supplementary table to allow the reader to easily re-order the rows based on values. I recommend this manuscript for publication.

We thank the reviewer for their recommendation of this manuscript for publication.

The table and associated p-values will be included for the now Figure 2A to ensure accessibility for readers.

Comments from Reviewer #3

Reviewer #3 (Remarks to the Author):

The authors have made substantial efforts to revise their manuscript and I am pleased for them that it has resulted in an additional finding.

We again thank the reviewer for their input in helping us revise the metabolomics portion and the manuscript and believe that it has substantially improved the article.

I have only one remaining issue: that of treatment of missing values. “Missing not at random” can refer to the distribution across samples, but it commonly is also used to describe the phenomenon that features with much lower average intensity are more likely to be missing than those with strong intensities. This can bias statistical results, especially where the missingness is more due to technical issues rather than true biological differences. The authors have not referred to this scenario in their analysis and therefore I don’t know if they have considered it.

The authors may therefore like to do a sanity check and just ensure that none of their reported important metabolites are affected by this issue. I don’t think that this should necessitate an additional round of review unless requested by the editor.

We thank the reviewer for sharing their thoughtful concern.

As recommended, we have now tested all excluded metabolites (both missing and those with low variance and found that none of them were statistically associated with allergies at 5 years, suggesting that they are truly missing at random and did not skew our analysis.

We have included both the result table and code used to run the analysis in the supplemental data of the manuscript.

Further, we have added the following sentence to the Methods section of our paper at Lines 510-512 indicating that all excluded metabolites were confirmed to not be significantly associated with an allergic disease:

“All excluded metabolites were confirmed to not be significantly associated with the presence of a 5-year allergy diagnosis (Supplementary Table 7).”